# Geostatistical assessment of warm-season precipitation observations in Korea based on the composite precipitation and satellite water vapor data

Sojung Park[1,3,4], Seon Ki Park[1,2,3,4], Jeung Whan Lee[5], and Yunho Park[5]

[1]Department of Climate and Energy Systems Engineering, Ewha Womans University, Seoul, Republic of Korea
[2]Department of Environmental Science and Engineering, Ewha Womans University, Seoul, Republic of Korea
[3]Severe Storm Research Center, Ewha Womans University, Seoul, Republic of Korea
[4]Center for Climate/Environment Change Prediction Research, Ewha Womans University, Seoul, Republic of Korea
[5]Korea Meteorological Administration, Seoul, Republic of Korea

**Correspondence:** S. K. Park (spark@ewha.ac.kr)

**Abstract.** Among the meteorological disasters, heavy rainfalls cause the second largest damage in Korea, following typhoons. To confront with the potential disasters due to heavy rainfalls, understanding the observational characteristics of precipitation is of utmost importance. In this study, we investigate the spatial and temporal characteristics of warm-season precipitation in Korea, according to the precipitation types, by conducting the geostatistical analyses such as autocorrelogram, Moran's $I$ and general $G$, on the composite (radar + station) precipitation data. The $e$-folding distance of precipitation ranges from 15 to 35 km, depending on the spatial distribution, rather than intensity, of precipitation, whereas the $e$-folding time ranges from 1 to 2 h. The directional analyses revealed that the warm-season precipitation systems in Korea, especially those with high precipitation amount, have high spatial autocorrelations in the southwest–northeast and west–east directions, in association with the frontal rainfalls, convection bands, etc. Furthermore, the cluster versus dispersion patterns and the hot versus cold spots are analyzed through Moran's $I$ and general $G$, respectively. Water vapor, represented by the brightness temperature, from three Himawari-8 water vapor bands also shows similar characteristics with precipitation but with strong spatial correlation over much longer distance ($\sim$100 km), possibly due to the continuity of water vapor. We found that, under the $e$-folding based standard, the current observation network of Korea is sufficient to capture the characteristics of most precipitation systems; however, under a strict standard (e.g., autocorrelation of 0.6), a higher-resolution observation network is essentially required — especially in local areas with frequent heavy rainfalls — depending on the directional features of precipitation systems. Establishing such an observation network based on the characteristics of precipitation enables us to improve monitoring/tracking/prediction skills of high-impact weather phenomena as well as to enhance the utilization of numerical weather prediction.

## 1 Introduction

Heavy rainfall causes one of the most serious meteorological disasters in Korea. Most precipitation systems in Korea are highly influenced by the East Asian summer monsoon, called *Changma* in Korean (Riyu et al., 2001; Kim et al., 2002; Lee et al., 2017b), and hence have high seasonal variation; they also show high spatial variation due to complex topographic

features (e.g., Ko et al., 2005; Chang and Kwon, 2007; Park and Lee, 2007; In et al., 2014). Since heavy rainfalls including typhoon cause serious damages to both human life and property, improving the their forecast accuracy is of utmost importance. Many studies have been done to improve the quantitative precipitation forecast using numerical weather prediction models and satellite or radar data (e.g., Fritsch et al., 1998; Lee et al., 2006; Kim and Oh, 2010; Feng and Kitzmiller, 2006; Yu et al., 2013).

In order to improve the forecast accuracy, it is essential to understand the spatial and temporal characteristics as well as the occurrence and development mechanisms of the precipitation systems.

The ground-based rainfall observation data, in Korea, are collected from the Automated Synoptic Observing Systems (ASOS), the Automatic Weather Stations (AWS), and the Automated Agriculture Observing System (AAOS). The observation density is about 67 km for ASOS and approximately 13 km by including AWS. The agrometeorological observation network

consists of 11 AAOS stations (Choi et al., 2015). Based on these precipitation observations, the mesoscale structures of precipitation as well as the hydrologic budgets in Korea have been extensively studied (e.g., Kim and Lee, 2006; Cassardo et al., 2009; Jeong et al., 2012; 2016; Jung and Lee, 2013; Lee et al., 2017a). Capturing the spatiotemporal features of precipitation systems from the observation networks is essential to runoff forecast, especially at the catchment scale and for the flooding cases (Volkmann et al., 2010). Given the limited resources, it is desirable to optimally design the observation networks, which

should be efficient to use the least amount of available instruments but sufficient to capture the precipitation characteristics — both amount and spatiotemporal variability. In other words, understanding temporal and spatial characteristics of precipitation is fundamental, and essentially required for establishment of the optimal observation networks (see Ciach and Krajewski, 2006; Volkmann et al., 2010). Moreover, the appropriateness of such observation networks, in analyzing and/or forecasting the precipitation systems, has seldom been assessed.

In this study, we investigate the spatial and temporal characteristics of precipitation in Korea using the geostatistical analysis methods — autocorrelogram, Moran's $I$, and general $G$ (e.g., Ciach and Krajewski, 2006; Schiemann et al., 2011; Emmanuel et al., 2012; Fu et al., 2014). Ciach and Krajewski (2006) estimated the spatial correlation functions in small-scale rainfall, based on raingauges covering an area of about 3 km by 3 km in Central Oklahoma. They showed a dependence of the correlogram parameters on the averaging time scale, large differences of the correlograms in the individual storms, and the ambiguities

in correlation estimates based on rainfall intensities. Bacchi and Kottegoda (1995) addressed the effects of weather systems, topography, and temporal/spatial scale of observation networks on spatial distribution of rainfall. Emmanuel et al. (2012) analyzed the characteristics of spatiotemporal variability by rainfall types using variogram.

In Korea, the spatial and temporal characteristics of precipitation have been reported in several previous studies (e.g., Chang and Kwon, 2007; Hong et al., 2006; Ha et al., 2007; In et al., 2014). Ha et al. (2007) found that the $e$-folding distance and time

in correlation coefficients of AWS precipitation ranged from 50 to 110 km and 1 to 2 h, respectively. In summer, precipitation showed localized features with relatively more precipitation mainly in the western part and southern coastal areas of Korea (Kim et al, 2005; Hong et al., 2006; In et al. 2014). In et al. (2014) also analyzed the degree of localization in summer precipitation, and noted a high locality of rainfalls, i.e., more frequent occurrence of short-lived local precipitation, in July and August. In addition to these studies, we provide further analyses on variation of spatial correlation associated with distribution

and intensity of precipitation.

Water vapor is also strongly linked to the precipitation system because not only it is the absolute source of precipitation (Eltahir and Bras, 1996; Park, 1999; Bretherton et al., 2004; Stohl et al., 2008; Gimeno et al., 2012) but also its transport in the lower troposphere provides diabatic heating and hence promotes development of convective precipitation systems (Tompkins, 2001; Trenberth and Stepaniak, 2003a, b; Smith and Yuter, 2010). Therefore, the amount and behavior of water vapor are the crucial factors in the precipitation system. For instance, a large amount of water vapor originated from the adjacent oceans is transported to the East Asian monsoon region through the large scale monsoon circulations. The origins of water vapor supply can also distinguish the anomalous rainfall patterns (e.g., Zhou and Yu, 2005). Furthermore, about 70 % of precipitation in extratropical cyclones is generated by water vapor already present when the storm formed in the atmosphere (Trenberth, 1999). Hence, we also conduct the spatial and temporal analysis of water vapor to enhance the understanding of precipitation characteristics over Korea and adjacent areas.

This study aims at classifying the precipitation types statistically based on the spatial distribution of precipitation, and identifying the spatial and temporal characteristics of warm-season precipitation in Korea according to the precipitation types via the geostatistical analysis methods, including autocorrelogram, Moran's $I$, and general $G$. Furthermore, we investigate the characteristics of water vapor over the Korean Peninsula associated with the precipitation types, using satellite data. Section 2 describes the data set used for the analyses, and Sect. 3 briefly introduces the geostatistical methods employed in this study. Results and discussions are provided in Sect. 4. Section 5 is devoted to conclusions.

## 2 Data description

We use the precipitation data from weather stations (see Fig. 1) to categorize the precipitation systems. We classify four different precipitation types statistically, based on two criteria: the portion of weather stations with precipitation (C1; in %), and the station average precipitation rate (C2; in mm h$^{-1}$). In order to determine the threshold values for classifying the precipitation types, we have conducted a preliminary statistical analysis on precipitation events in the period of 2011–2015 (not shown). As the precipitation events occur in a given time period and space interval, our precipitation data are assumed to follow the Poisson distribution, which represents a probability situation of a large number of observation with a small probability of occurrence. Many studies developed the Poisson distribution models to estimate rainfall and to cluster the rainfall systems (e.g., Rodriguez-Iturbe et al., 1987; Lee et al., 2014; Barton et al., 2016; Ritschel et al., 2017). We have chosen the threshold values when the cumulative percentage of precipitation events for each criterion (i.e., C1 and C2) reached approximately 80 %; thus obtaining the threshold values of 20 % for C1 and 3 mm h$^{-1}$ for C2, respectively. Our preliminary statistical analysis showed that, in general, most precipitation events occur over small areas and precipitation events with high intensity rarely occur over large areas. The locality of precipitation appeared higher as the precipitation intensity were higher, in accordance with Nam et al. (2014). In particular, precipitation systems with the highest intensity ($\geq 10$ mm h$^{-1}$) were mostly confined to a small area with the number of stations less than 10 % of total weather stations. This implies that the locality feature of precipitation systems may depend on the threshold value in precipitation intensity.

Based on these criteria, we define four different precipitation types, as shown in Table 1: 1) Low Precipitation at a Few Points (LPFP) for $C1 < 20$ % and $C2 < 3$ mm h$^{-1}$; 2) Low Precipitation at Many Points (LPMP) for $C1 \geq 20$ % and $C2 < 3$ mm h$^{-1}$; 3) High Precipitation at a Few Points (HPFP) for $C1 < 20$ % and $C2 \geq 3$ mm h$^{-1}$; and 4) High Precipitation at Many Points (HPMP) for $C1 \geq 20$ % and $C2 \geq 3$ mm h$^{-1}$. We practically exclude the LPFP type in our analyses, i.e., the case with $C1 < 20$ % and $C2 < 3$ mm h$^{-1}$, for it may be less effective.

The Korea Meteorological Administration (KMA) has produced the composite precipitation data over Korea using the data from radars, weather stations and satellites, through the following steps (see Hwang et al., 2015): 1) remove non-precipitation echoes from the radar data using the satellite cloud type data; 2) calculate the difference between the station precipitation and the radar estimated precipitation; 3) perform the objective analysis on the precipitation difference field and on the station precipitation data; 4) correct the bias using the objectively-analyzed difference field; and 5) combine the corrected radar-estimated precipitation data and the objectively-analyzed station precipitation data to produce the composite precipitation data (in mm h$^{-1}$). In order to analyze the precipitation systems with the evenly distributed high-resolution data, we used these composite precipitation data, which cover 1153 km × 1441 km over the Korean Peninsula, with a grid size of 1 km and a time resolution of 1 h. Geostatistical analyses are conducted using the composite precipitation data sets from April to October in a period of 2013–2015 to investigate the spatial and temporal characteristics of warm-season rainfall.

We also used the Himawari-8 water vapor bands to investigate the distribution feature of water vapor in the lower to upper atmosphere in association with precipitation. The water vapor bands are strongly linked with the properties of moisture, thermodynamics and dynamics of the troposphere. As they are sensitive to the moisture and temperature profiles in the radiation path, they can provide information for a wide range of atmospheric processes (Georgiev et al., 2016), including atmospheric motions such as the upper-level lows, jet stream, and blocking. Himawari-8, operated by the Japan Meteorological Agency since 7 July 2015, carries a new unit called the Advanced Himawari Imager (AHI) with higher radiometric, spectral, and spatial resolution in the geostationary orbit (Bessho et al., 2016). The AHI has 16 observation bands — three visible (0.47, 0.51, 0.64 $\mu$m), three near-infrared (0.86, 1.6, 2.3 $\mu$m), and ten infrared (3.9, 6.2, 6.9, 7.3, 8.6, 9.6, 10.4, 11.2, 12.4, 13.3 $\mu$m). Among them the water vapor absorption bands — 8 (6.2 $\mu$m), 9 (6.9 $\mu$m), and 10 (7.3 $\mu$m) — represent upper-, mid-, and mid-/lower-level water vapor, respectively. The calibration of the Himawari-8 water vapor bands is accurate to within 0.2 K by validating an approach developed under the Global Space-based Intercalibration System (GSICS) project with hyper-spectral infrared sounders (e.g., Okuyama et al., 2015; Bessho et al., 2016). In this study, we used the scan area for Full Disk by the AHI that covers the East Asia region: the Full Disk images are taken every 10 min with a 2 km spatial resolution. The analyses are conducted on the data of the hour over the domain of 120.132−134.243°E and 30.436−44.068°N (600 × 770 grids), similar to the area of the precipitation data.

## 3  Geostatistical methods

In this study, we use several methods to analyze the spatial characteristics — the spatial autocorrelation, Moran's $I$, and general $G$. The spatial autocorrelation shows how the precipitation observations at a certain distance are correlated with each other.

Thus, it is possible to determine the degree of spatial scale, or the effective influence range, of a single precipitation system or associated precipitation systems. Through Moran's $I$, we can figure out the property of spatial correlations, especially distinguishing between the cluster and dispersion patterns/areas of precipitation. In other words, Moran's $I$ identifies local cluster areas of strong precipitation, large areas of precipitation, or precipitation boundaries. General $G$ indicates whether the cluster region has high or low precipitation.

### 3.1 Spatial autocorrelation

Autocorrelation in spatial analysis implies that a variable is correlated with itself at different points. Simply speaking, pairs of precipitation data close to each other likely to have more similar values than those far apart from each other. When the precipitation data are spatially autocorrelated, precipitation at one place is related to that at another place, and they are not independent to each other. The autocorrelation coefficient can be obtained once the autocovariance has been calculated. The autocovariance indicates the degree of similarity of a variable itself at a certain distance. Generally, the autocovariance is greater as the separation distance decreases, becomes smaller as the distance increases, and does not show any tendency (e.g., converge) after a certain distance. For the one-dimensional data, the autocovariance ($cov$) in terms of the grid distance (or lag; $k$) between the data pairs, $X = (x_1, x_2, \cdots, x_{n-k})$ and $X' = (x_{1+k}, x_{2+k}, \cdots, x_n)$, is defined as:

$$
\begin{aligned}
cov(X, X') &= E[(X - \mu_X)(X' - \mu_{X'})] \\
&= E(XX') - E(X)E(X') \\
&= \mu_{XX'} - \mu_X \mu_{X'}.
\end{aligned}
\tag{1}
$$

Here $E(XX')$, $E(X)$ and $E(X')$ are the means of sub-datasets (i.e., $XX'$, $X$ and $X'$, respectively), and are calculated as the followings:

$$
E(XX') = \frac{1}{n-k} \sum_{i=1}^{n-k} x_i x_{i+k} = \mu_{XX'},
\tag{2}
$$

$$
E(X) = \frac{1}{n-k} \sum_{i=1}^{n-k} x_i = \mu_X
$$

and

$$
E(X') = \frac{1}{n-k} \sum_{i=1}^{n-k} x_{i+k} = \mu_{X'},
$$

where $n$ is the number of total points and $i$ is the grid point index. The lag index, $k$, depicts the grid distance between two point values, $x_i$ and $x_{i+k}$. The autocorrelation coefficient ($\rho$) is defined as:

$$
\rho(X, X') = \frac{cov(X, X')}{\sqrt{var(X)var(X')}} = \frac{\mu_{XX'} - \mu_X \mu_{X'}}{\sqrt{var(X)var(X')}},
\tag{3}
$$

where $var(X)$ means the variance of $X$ and can be regarded as the covariance of $X$ with itself, i.e., $var(X) = cov(X, X)$. At the zero separation distance, the autocovariance has the same value as the variance and the autocorrelation coefficient becomes

unity. For the two-dimensional data, the autocorrelation coefficient of all data pairs within the separation distance is obtained by using the relaxed separation distance, and by following the same calculation principle as in the one-dimensional data. In order to compute the relaxed separation distance, the distance between the data pairs is rounded.

## 3.2  Moran's $I$

Moran's $I$ is a statistical index for measuring the similarity of neighboring data (Moran, 1948). It is obtained by comparing the values of the target and neighboring feature pairs with the average of the entire targets:

$$I = \frac{n \sum_i \sum_j w_{ij}(x_i - \overline{x})(x_j - \overline{x})}{\sum_i \sum_j w_{ij} \sum_i (x_i - \overline{x})^2}, \tag{4}$$

where $n$ is the number of the entire target observations, $x_i$ is the variable value at the $i$-th location, $\overline{x}$ is the mean of the entire targets. Here, $w_{ij}$ is the spatial weight of the link between $i$ and $j$, which is defined by the inverse distance weight, i.e., $w_{ij} = 1/d_{ij}$ with $d_{ij}$ representing the distance between grids $i$ and $j$. Moran's $I$ is ranged from $-1$ to $1$ with the following interpretation: 1) the positive value means that the variable of interest and its spatial lags are positively autocorrelated, defining a *cluster* pattern; 2) the negative value indicates spatially negative autocorrelation, defining a *dispersion* pattern; and 3) the zero value represents a *random* spatial pattern. Equation (4) denotes the global Moran's $I$ that identifies the pattern — either cluster or dispersion — but with no information on the magnitude of variables. In addition, the local Moran's $I$ ($I_i$), showing the location of cluster/dispersion, is computed as the following (Anselin, 1995):

$$I_i = \frac{(x_i - \overline{x})}{\sum_{k=1}^n (x_k - \overline{x})^2/(n-1)} \sum_{j=1}^n w_{ij}(x_j - \overline{x}). \tag{5}$$

$I_i$ is interpreted the same as $I$ but with no specific range. The Z-score ($Z$) can be analyzed together to confirm the significance of $I_i$, and is calculated as:

$$Z(I_i) = \frac{I_i - E(I_i)}{SD(I_i)}, \tag{6}$$

where $E(I_i) = -\sum_j w_{ij}/(n-1)$ is the expected value of $I_i$ if random, and $SD(I_i)$ is the standard deviation of $I_i$. The Z-score suggests whether we can reject the null hypothesis or not. In this case, the null hypothesis states "there is no spatial clustering". To determine if the Z-score is statistically significant, we compare it to the range of values for a particular confidence level. For example, at the significance level of 0.1, 0.05, 0.01, the absolute value of Z-score would have to be greater than 1.65, 1.96, 2.57, respectively, to be statistically significant, and the clusters are not created by chance.

## 3.3  General $G$

We can identify the cluster patterns through Moran's $I$; however, it does not indicate whether the clustered variable values are low or high. In addition to Moran's $I$, it is necessary to determine whether the properties of the cluster are hot spots (i.e., high values) or cold spots (i.e., low values) by using Getis-Ord general $G$ (Getis and Ord, 1992; hereafter, referred to as general $G$). The global general $G$, like the global Moran's $I$, shows the cluster characteristics for the entire study area as a statistical index.

Therefore, we use the local general $G$ which can represent locations where features with either high or low values are spatially clustered. The following equation calculates the local general $G$ ($G_i(d)$):

$$G_i(d) = \frac{\sum_j w_{ij}(d)x_j}{\sum_j x_j}, \quad j \neq i, \tag{7}$$

where $d$ is the distance between the target feature and the neighboring feature, and $w_{ij}$ is the same spatial weight (i.e., the inverse distance weight) used for calculating Moran's $I$ as in Eqs. (4) and (5). To be a statistically significant hot spot or cold spot, the Z-score of general $G$ can be calculated. Ord and Getis (1995) redefined $G_i(d)$, which can be considered as the Z-score ($Z(G_i)$), as:

$$Z(G_i) = \frac{G_i - E(G_i)}{SD(G_i)}, \quad j \neq i, \tag{8}$$

where $E(G_i) = \sum_j w_{ij}(d)/(n-1)$ is the expected value if random, and $SD(G_i)$ is the standard deviation of $G_i$. High positive values indicate the possibility of a local cluster of high values, whereas negative values indicate a similar cluster of low values. An absolute value of the Z-score would have to be greater than 1.65, 1.96, 2.57 to be statistically significant at a significance level of 0.1, 0.05, 0.01, respectively.

## 4 Results and discussions

### 4.1 Spatial autocorrelation

#### 4.1.1 Precipitation

The general characteristics of the spatial autocorrelation structure in precipitation for each type are shown in Fig. 2. We have calculated the correlation coefficients for all points in the range from 0 to 100 km. In order to compare the characteristics between different precipitation types, we defined the $e$-folding value as the threshold. The $e$-folding distance means the separation distance when autocorrelation becomes $e^{-1}$ (i.e., 0.3674). In other words, it refers to the spatial scale of precipitation that autocorrelation decreases to $\rho = e^{-1}$ from the value at the zero separation distance (i.e., $\rho = 1$). The $e$-folding value, under the assumption that atmospheric variables decrease exponentially with time and distance, has often been used to estimate both the temporal and spatial structure of precipitation (e.g., Skøien et al, 2003; Ha et al., 2007; Kursinski and Mullen, 2008; Zeweldi and Gebremichael, 2009; In et al., 2014). As classified in Table 1, the LPMP and HPMP cases (i.e., C1 is high, regardless of C2) show the $e$-folding distance at ∼35 km. On the other hand, the HPFP cases (i.e., C1 is low) depict the $e$-folding distance at ∼15 km.

This result can be expected from the classification of precipitation types: precipitation with smaller spatial scale brings about more rapid decrease in the spatial correlation, thus showing smaller correlation at the $e$-folding distance. Ha et al. (2007) estimated the $e$-folding distance of warm-season precipitation as 50–110 km, and In et al. (2014) pointed out that the $e$-folding distance varied depending on months and regions with the minimum being 40 km. Our results show shorter $e$-folding distances than those from the previous studies: it is ascribed to exclusion of the LPFP cases (see Table 1). Nam et al. (2014) examined the

relationship between precipitation intensity and spatial autocorrelation, and noticed that the latter became smaller as the former appeared stronger, thus depicting a larger locality in precipitation. They found that the $e$-folding distance was 14 km for the precipitation intensity in excess of 30 mm h$^{-1}$, whereas it was about 40 km for the intensity greater than 5 mm h$^{-1}$. However, our results indicate that the spatial autocorrelations of HPMP and LPMP are in similar shape. Generally, the spatial locality is proportional to precipitation intensity; however, strictly speaking, the number of points with precipitation has a greater effect on spatial characteristics than the precipitation intensity.

In Fig. 3, in order to identify the spatial characteristics of precipitation for each case, we show a histogram depicting the number of cases according to the $e$-folding distance. We found that the $e$-folding distance of the mode (the most frequent value) was 15 km for HPFP, 30 km for LPMP, and 34.7 km for HPMP, respectively. It is evident that, as shown in Fig. 2, the $e$-folding distance depends on the precipitation spatial scale rather than the precipitation intensity.

We additionally analyzed the directional features of precipitation distribution over the Korean Peninsula in order to know the spatial structures of precipitation systems (see Figs. 4 and 5). We classified the direction range from 0° to 180° with an interval of 45°, considering the symmetry. Here the direction is a measure of the angle from the origin–east axis, whose magnitude increases counterclockwise (see the radar chart in Fig. 5). Figure 4 shows the averaged directional autocorrelation for different precipitation types. The autocorrelation curves show similar shape and property in HPMP (Fig. 4a) and HPFP (Fig. 4c), which depict two distinct directional characteristics: the precipitation structures have similar spatial features in directional pair of 45° and 0° and of 135° and 90°. For HPFP where the precipitation spatial scale is small, the autocorrelations of all directions decrease more rapidly than those for LPMP and HPMP, resulting in a smaller gap between the two distinct curves compared to HPMP. For LPMP, i.e., low precipitation rate with a large spatial scale, the spatial correlation is the highest with the precipitation axis of 45° (i.e., southwest–northeast), whereas it is the lowest with that of 135° (northwest–southeast).

Figure 5 shows a radar chart representing the directional $e$-folding distance at the mode in the directional histogram of case numbers (not shown). In general, the directional $e$-folding distances (and hence the spatial correlations) are larger for the cases with large spatial scale (i.e., LPMP and HPMP) than the cases with small spatial scale (i.e., HPFP). It is noteworthy that the $e$-folding distances in LPMP have no characteristic directionality (i.e., isotropic), indicating that the precipitation systems in LPMP are mostly related to both the migratory cyclones that move into Korea from various directions and the cloud clusters where moderate convective systems in large area are closely connected to each other. On the other hand, the cases with high precipitation intensity, i.e., HPFP and HPMP, both show strong directionality (i.e., anisotropic) with large $e$-folding distances along the axes of 45° and 0° (i.e., the southwest–northeast and west–east directions, respectively). They are considered to be strongly linked to the frontal rainfalls, convection bands, etc. (e.g., Sun and Lee, 2002; Kim and Lee, 2006; Lee et al., 2008; Jeong et al., 2012, 2016; Jung and Lee, 2013).

As we have used the 3-year data, the spatial autocorrelations here are considered to include the temporal features implicitly. All the weather systems evolve in time (i.e., develop, mature and decay) and move in space during their life cycles. The fact that precipitation systems have high correlation along a specific direction implies that those systems in that direction, even in the far distances, have similar/common structures or are originated from the same weather system. For instance, the meteorological systems with strong directionality include a squall line or a frontal system in which several thunderstorms banded together,

and a multicell cluster that comprises a series of individual storm at a different stage of life cycle with the same movement direction. For the multicell cluster, new cells form along the upwind edge of the cluster, and decaying cells are found along the downwind side with mature cells located at the center; thus it includes evolutions in both space and time.

In terms of the summertime heavy precipitation systems over the Korean Peninsula, Lee and Kim (2007) classified them into four major types such as isolated thunderstorms, convection bands, squall lines and cloud clusters, whereas Song and Sohn (2015) classified them into two types: the cold type characterized by an eastward moving cloud system with an oval shape, and the warm type with a comparatively wide spatial distribution over an area extending from the southwest to northeast. In our case, the precipitation events in HPFP may correspond to the isolated thunderstorms or the cold-type heavy rainfalls while those in HPMP to convection bands, squall lines, cloud clusters or the warm-type heavy rainfalls. On the other hand, Ha et al. (2007) investigated the monthly characteristics in the directional features of rainfall from May to September, and reported that the cell type was dominant in May and September while the southwest–northeast tilted band type was ruling in June–August. They addressed that the band type is mainly due to frontal rainfall during the rainy season. In our results, most cases of HPMP and HPFP occurred in July and August (see Table 2); thus, indicating that the warm-season precipitation systems in Ha et al. (2007) are mostly related to the long-lived (HPMP) and short-lived (HPFP) band type heavy rainfalls along the monsoon frontal system.

### 4.1.2  Himawari-8 water vapor bands

Water vapor is the core element and driver of the precipitation development through dynamical processes (e.g., advection and convection) and physical processes (e.g., evaporation and condensation). For example, the East Asian monsoon starts when a huge amount of water vapor from the adjacent ocean is transported to the monsoon region by the large scale atmospheric circulation. Thus, the spatial analysis of water vapor will contribute to further understanding of the spatial patterns of precipitation. Many studies have been carried out on the relationship between satellite water vapor and extratropical/tropical cyclones and storms (e.g., Velden, 1987; Milford and Dugdale, 1990; Krennert and Zwatz-Meise, 2003; De Haan et al., 2004; Rabin et al., 2004; Mukhopadhyay et al., 2005; Chosh et al., 2008).

By analyzing the water vapor imagery, we can detect not only the horizontal distribution of tropospheric water vapor but also the dynamical behavior of atmospheric flow such as the middle and upper troughs, vortexes, and jet streams, even in the absence of clouds. Therefore, analyzing the spatial characteristics of tropospheric water vapor is essential to improve the understanding of precipitation systems. In particular, we are focusing on the extent and direction of spatial correlation of water vapor flowing into the Korean Peninsula in association with the precipitation types.

In this study, we analyze the brightness temperatures from the Himawari-8 water vapor bands to characterize the lower to upper atmosphere related to the precipitation systems. A humid atmosphere absorbs more longwave radiation from the Earth, resulting in a lower brightness temperature. On the other hand, a dry atmosphere absorbs less longwave radiation, bringing about a higher brightness temperature. Although we cannot directly quantify the amount of water vapor through the water vapor imager, we can sufficiently recognize the spatial distribution of water vapor. Moreover, using two water vapor bands (i.e., 6.2 and 7.3 $\mu$m), we can clearly identify the moisture boundaries at the zone between the warm/moist and cold/dry side of

the jet/wind maximums at two different levels in the troposphere (Georgiev et al., 2016). The spatial analyses were performed with the mixed images of clouds and water vapor because it was hard to distinguish between clouds and water vapor without a cloud detection algorithm. Since both water vapor and clouds are strongly linked to precipitation as its sources, analysis of the mixed variables from the satellite data would not make a serious problem in understanding and relating to the precipitation

systems. As we focus on the spatial distribution of water vapor when precipitation occurs, we analyze water vapor for each precipitation type.

The general characteristics of the spatial autocorrelation structure in water vapor bands for each precipitation type are shown in Fig. 6. The patterns of spatial autocorrelations of water vapor bands, for three precipitation types, are generally similar to those of precipitation; however, the separation distances extend much longer (cf. Fig. 2). Even at a separation distance of

10 100 km, the spatial correlation is higher than 0.6. This strong spatial correlation is considered to be due to the continuity of water vapor. The largest spatial autocorrelation appears in LPMP, followed by HPMP and HPFP. The LPMP depicts similar correlation coefficients regardless of bands and hence heights of atmospheric layers. This implies that the spatial structure of water vapor bands is similar in the vertical. On the other hand, the spatial correlations in HPMP and HPFP become smaller along the downward path, from band 8 to band 10.

In Fig. 7, the brightness temperature of water vapor bands characterizes directionality in the spatial correlations. In general, for all precipitation types and water vapor bands, the highest spatial correlations appear in the axis of $45°$ (southwest-northeast) while the lowest in the axis of $135°$ (northwest-southeast), as in precipitation (cf. Fig. 4). In LPMP, the spatial correlations in the axes of $0°$ and $90°$ coincide even at longer distances. We also notice that, for a given precipitation type and separation distance, the spatial autocorrelation of water vapor becomes larger along the upward path, from band 10 to band 8. Figure

8 shows the directional separation distances at the mode in the directional histogram of case numbers (not shown) for each precipitation type with the autocorrelation coefficients of 0.7 (blue), 0.8 (black), and 0.9 (grey), respectively. The separation distances generally show high values mainly in the axis of $45°$, similar to the directional features of precipitation (cf. Fig. 5). Note that, in LPMP, the brightness temperature shows a large separation distance (i.e., high spatial correlation) in the axis of $45°$ while precipitation depicts no directionality (cf. Fig. 5). This is possible because water vapor can form as a band type,

e.g., over a stationary front but precipitation can occur in cell type due to local features of convection over the same front. In addition, as the autocorrelation coefficient decreases, the directionality tends to disappear even though the autocorrelation coefficient is as large as 0.7. As in Fig. 7, we note that for a given precipitation type and spatial autocorrelation, the separation distance of water vapor becomes larger along the upward path. This is related to the stronger wind at upper atmosphere that carries the characteristic feature of water vapor further downstream.

Through the satellite water vapor analyses, we found both similarity and dissimilarity in spatial correlations between water vapor and precipitation. Similar to the results of precipitation analyses, the spatial autocorrelations of water vapor for HPFP decreased more rapidly than those for HPMP and LPMP (cf. Figs. 2 and 6). Water vapor and precipitation showed additional similarity in terms of characteristic directionality in spatial autocorrelations (cf. Figs. 4 and 7, and Figs. 5 and 8). However, both fields showed significant dissimilarity in the separation distances, i.e., the spatial scales (cf. Figs. 2 and 6). Note that water

vapor makes phase changes and the conversion from water vapor to precipitation includes a bunch of nonlinear processes;

thus it is not surprising to see such dissimilarity. Further studies on the relationship between precipitation systems and satellite water vapor in Korea are essentially required.

## 4.2 Moran's $I$ and general $G$

Moran's $I$ represents the spatial distribution pattern, i.e., cluster or dispersion: the cluster (dispersion) patterns appear in the areas with positive (negative) spatial autocorrelation. General $G$ determines whether the cluster area is a hot or cold spot: a hot (cold) spot means an area with high (low) values and hence high (low) precipitation amount. For all precipitation cases in this study, the values of global Moran's $I$ showed positive values meaning that clusters were detected, in the domain-average sense. Through the local Moran's $I$ ($I_i$), we can define the similarity between the target feature and its neighbors, and check the spatial outliers as well (Lalor and Zhang, 2001; McGrath and Zhang, 2003). For example, with a negative $I_i$, the target feature may have high precipitation while its surroundings have low precipitation, and vice versa. Conversely, with a positive $I_i$, the target feature has similar amount of precipitation as its neighbors. Thus, the local Moran's $I$ can detect strong precipitation over local clusters, precipitation boundaries, or precipitation over large areas. As Moran's $I$ cannot distinguish between low and high precipitation, we employed the local general $G$ to identify the hot/cold spots (i.e., high/low precipitation areas, respectively). In order to calculate the indices, we selected a sub-domain for each precipitation case where the precipitation values were extracted at an interval of 5 km. An inverse distance weighting function is used as the spatial weight.

We present the analysis results for three cases — one from each precipitation type (see Figs. 9–11). For the selected cases, the values of global Moran's $I$ were all positive: 0.1207, 0.1711 and 0.2357 for the cases of LPMP, HPMP and HPFP, respectively, with a higher value representing a stronger clustering. This indicates that all three cases show the cluster patterns. Figure 9 shows an LPMP case, occurred at 05 Korean Standard Time (KST) 25 August 2015, in association with a tropical cyclone. Typhoon Goni (2015) approached from Japan and brought rainfall all over Korea (Fig. 9a). In particular, heavy precipitation systems, with more than 30 mm h$^{-1}$, developed around the mountain range along the east-central coast of the Korean Peninsula. In Fig. 9b, the southeastern region is identified as hot spots (areas of high precipitation) while the southwestern region, over both land and sea, as cold spots (areas of low precipitation). Figure 9c depicts several precipitation systems with maximum intensity (i.e., precipitation rate; see Fig. 9a) in the cluster area (marked by the X symbols). These systems show the highest local Moran's $I$ with the spatial scale of less than 30 km. The cluster patterns were statistically significant at a significance level of 0.01 (Fig. 9d). In Fig. 10, we illustrate a heavy rainfall case in the category of HPMP, which occurred at 17 KST 27 May 2013 as a migratory cyclone approached from the west, passed over the Korean Peninsula slowly, and brought rainfall throughout the country, mainly in the western and southern regions (Fig. 10a). Such precipitation areas matched well with the hot spots in Fig. 10b. The map of local Moran's $I$ shows that a precipitation system with a strong cluster feature has developed, over the southwestern sea of the Korean Peninsula, along with the successive cluster patterns in a line type (Fig. 10c), with a significance level of 0.01 (Fig. 10d). Figure 11 shows an HPFP case, occurred at 05 KST 24 October 2015. The central-western region of the Korean Peninsula received rainfall of more than 8 mm h$^{-1}$ (Fig. 11a): the hot spots appeared in the precipitation area while the cold spots were located at the region of relatively low precipitation (Fig. 11b). Figure 11c

illustrates an intense cluster pattern (dark red) over the relatively high rainfall area, and a weaker cluster pattern over the cold spot, with a significance level of about 0.01 (Fig. 11d).

Overall, the spatial structures of the selected precipitation systems are well presented by the analyses of Moran's $I$ and general $G$. The overall spatial pattern (cluster vs. dispersion) of a precipitation system is characterized by the global Moran's $I$, whereas the spatial distributions of clustered areas are represented by the local Moran's $I$. The overall regional features of precipitation amount (i.e., hot vs. cold spot) are captured by the local general $G$. For example, precipitation in the HPFP case, relevant to convective clusters, appeared over a small local area (Fig. 11a) but the value of global Moran's $I$ was much higher (0.2357) than the others (0.1207 and 0.1711), implying a stronger cluster pattern; however, considering that the global Moran's $I$ is a domain-averaged value, this high value may be due to less dispersion (negative correlation) areas. In fact, the HPFP case shows a relatively weaker dispersion area (Fig. 11c), compared to the LPMP and HPMP cases (Figs. 9c and 10c, respectively). It is notable that the cluster pattern shows different spatial characteristics depending on the spatial scale and distribution feature of precipitation systems: several strong local clusters develop, especially over areas of high precipitation, when precipitation occurs over a large region (Figs. 9 and 10), whereas weaker clusters develop when precipitation occurs over a small region (Fig. 11). Through the integrated interpretation of these indices, we discovered some important spatial features of the precipitation systems: 1) the area where the cluster pattern overlaps with the hot spots is characterized by heavy rainfall (e.g., the X-marked or dark red places in Figs. 9c and 10c); 2) the area where the dispersion pattern concurs with the cold spots is distinguished by a relatively high precipitation surrounded by low precipitation (e.g., the rain cells denoted by arrows in Figs. 9c and 10c); 3) the precipitation gets weaker as it moves away from the strong cluster area, that is, the cluster pattern appears at the center and the dispersion pattern appears around; and 4) the secondary cluster pattern in a precipitation system over a wide area indicates clusters of low-intensity precipitation.

## 4.3 Temporal autocorrelation

We additionally calculated the temporal autocorrelation coefficients of precipitation and water vapor bands for each precipitation type. For precipitation, HPFP represents the shortest $e$-folding time, followed by LPMP and HPMP (Fig. 12). This order in the $e$-folding time is the same as in the $e$-folding distance: this implies that, for the precipitation types classified here, the temporal correlation as well as the spatial correlation appears greater in the precipitation systems over a larger area than those over a smaller area. The range of the $e$-folding time for all precipitation types is 1–2 hours. The averaged temporal autocorrelations of water vapor bands show similar features to those of precipitation for different precipitation types, though the correlations decrease slowly over time (Fig. 13). In addition, compared with the spatial correlations, the temporal correlations do not show significant dependence on the height of atmospheric layer.

Through this temporal correlation analyses, we noticed that the $e$-folding time is in a short range (1–2 hours), and its difference among different precipitation types is only about 30 min. This short $e$-folding time scale for all precipitation types implies that the precipitation systems affecting Korea in the warm season are mostly characterized by convective-type precipitation, from either isolated storm cells or clustered bands, at least for the given analysis period. The $e$-folding time scale suggests an adequate time interval for data collection and analysis for capturing the detailed structure of and better forecasting of precipi-

tation systems. It also implies a proper time interval of incorporating observations in the operational data assimilation system, for more accurate numerical forecasting of the precipitation systems.

Moreover, Ha et al. (2007) reported that the $e$-folding time of precipitation in Korea is 1–2 h regardless of months (from May to September), and that the monthly difference of the $e$-folding time is approximately 30–40 min. Therefore, we can conclude that the typical $e$-folding time of the warm-season precipitation systems in Korea is 1–2 h, regardless of the precipitation types and months. This conclusion is based on the analyses of the hourly precipitation data: we may find different temporal characteristics for different precipitation types using a data set of shorter interval (e.g., 10 min). In terms of the satellite water vapor data, a further study is also necessary to investigate the relationship between water vapor transport and precipitation, regarding the temporal scale as well as the spatial scale, with more detailed analyses.

## 5   Conclusions

Heavy rainfall causes many casualties as well as property damage, and thus its accurate forecast is very important. To improve the forecast accuracy, it is necessary to understand the characteristics of the precipitation systems through analyses of data from observation and/or numerical modeling. Furthermore, a well-designed observation network is essentially required to capture the characteristic features of precipitation systems.

In this study, we identify the characteristics of warm-season precipitation systems in Korea, via the geostatistical analyses on the composite precipitation and satellite water vapor data. We have classified the precipitation cases into four types, based on the average rainfall amount per point and the ratio of the points with precipitation (i.e., spatial scale): 1) Low Precipitation at a Few Points (LPFP); 2) Low Precipitation at Many Points (LPMP); 3) High Precipitation at a Few Points (HPFP); and 4) High Precipitation at Many Points (HPMP), among which the LPFP cases are excluded from the analysis.

We have conducted analyses of the spatial and temporal autocorrelations of precipitation and water vapor for the precipitation cases of each classification type, and found that the $e$-folding distance of precipitation ranges between 15 and 35 km while the $e$-folding time ranges between 1 and 2 h. The $e$-folding distance is as short as 15 km for an HPFP case (e.g., a local shower), but as large as  35 km for the cases of both LPMP and HPMP, implying that the spatial correlation becomes larger as precipitation occurs over a wider area and is less affected by the precipitation amount. We also noted that the spatial autocorrelations have characteristic directionality as the followings: 1) precipitation systems with high precipitation amount (i.e., HPFP and HPMP) have high spatial autocorrelations in the southwest–northeast and west–east directions, mainly associated with frontal rainfalls during the monsoon season; and 2) those with low precipitation amount (i.e., LPMP) have no directionality, pertaining to both the migratory cyclones moving in various directions and the moderate convective systems in the cloud clusters. Other spatial characteristics were identified as well, including cluster versus dispersion patterns, hot versus cold spots, strong precipitation over local clusters, precipitation boundaries, precipitation over a large area, and so forth. In general, heavy rainfalls appear over the hot spots with the cluster pattern while an isolated high precipitation system, surrounded by low precipitation, occurs over the cold spots with the dispersion pattern.

The Advanced Himawari Imager (AHI), on board Himawari-8, has three water vapor bands — 8 (6.2 $\mu$m), 9 (6.9 $\mu$m) and 10 (7.3 $\mu$m) — representing lower atmospheric layer with higher wavelength. Since water vapor has continuity in time and space, the autocorrelation coefficients of water vapor bands drop more slowly than those of the composite precipitation. However, the comparative analyses according to the precipitation type reveal that the spatial correlations of water vapor bands behave similarly to those of precipitation: the correlations of water vapor bands are also large (small) for precipitation systems occurred over a large (small) area.

We can also assess the current observational network in Korea, focusing on the warm-season precipitation systems, via the geostatistical analyses. Currently, the precipitation observation network in Korea has a spatial resolution of ~13 km, distributing the analysis in an interval of 1 h. For a standard based on the $e$-folding values, such observation network is capable of deriving the spatiotemporal characteristics of the precipitation systems in all three types — LPMP, HPMP and HPFP. However, if we apply a strict standard (e.g., autocorrelation of 0.6), the separation distance and time for the HPFP cases fall down to ~6 km and ~35 min, respectively. This implies that, regarding the spatial resolution, the current observation network can hardly capture the characteristic features of the localized precipitation systems; the temporal resolution might be sufficiently high because the composite precipitation data are produced at every 10 min. Therefore, for more accurate analyses of the precipitation systems with a higher correlation, a denser observation network should be considered in local areas where a heavy rainfall often occurs. This study has enabled us not only to explore the characteristics of precipitation systems in Korea but also to suggest the criteria for evaluating the observation network.

We note that the spatial distribution and movement features of the precipitation systems are strongly affected by terrain, but the orographic effect is not fully assessed in this study. Bacchi and Kottegoda (1995) demonstrated the orographic effect of the Alps mountain range on the spatial distribution of the rainfalls and addressed the directional characteristics of the spatial correlations. Other studies also confirmed the direct and induced effect of mountain on spatial distribution of rainfall (e.g., Mass, 1981; Carruthers and Choularton, 1983; Barros and Kuligowski, 1997; Park and Lee, 2007). As the Korean Peninsula is represented by a highly complex terrain with many mountains, the spatial properties of precipitation systems will be distinctively affected. It is essential to conduct a further research about the orographic effect on the spatiotemporal characteristics of precipitation, specifically over different slope sides of various mountain ranges in Korea. We additionally suggest a further study to analyze the autocorrelations by considering the storm cell movement and the occurrences of initial precipitation and maximum rainfall intensity.

*Data availability.* The composite precipitation and Himawari-8 data are available from the Korea Meteorological Administration database: http://203.247.66.28/ (instructed in Korean), in which a user registration is required. The Himawari-8 data is also disseminated via the HimawariCast from the Japan Meteorological Agency at http://www.data.jma.go.jp/mscweb/en/himawari89/himawari_cast/himawari_cast.html.

*Acknowledgements.* This work is supported by the Korea Meteorological Administration (KMA) Research and Development Program under Grant KMIPA, mainly through KMIPA2016-1010 and partly through KMI2018-06710. We acknowledge KMA for supplying the composite precipitation and the Himawari-8 water vapor bands data.

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

**Table 1.** Classification of precipitation types with the number of events (in bold).

| | | The portion of weather stations with precipitation (C1) | |
|---|---|---|---|
| | | < 20 % | ≥ 20 % |
| The station average precipitation rate (C2) | < 3 mm h$^{-1}$ | Low Precipitation at a Few Points (LPFP) **5858** | Low Precipitation at Many Points (LPMP) **1157** |
| | ≥ 3 mm h$^{-1}$ | High Precipitation at a Few Points (HPFP) **980** | High Precipitation at Many Points (HPMP) **594** |

**Table 2.** Monthly occurrences of precipitation systems according to the precipitation types from April to October in the period of 2013–2015.

| | LPMP | HPMP | HPFP |
|---|---|---|---|
| APR | 277 | 31 | 21 |
| MAY | 106 | 54 | 37 |
| JUN | 130 | 50 | 186 |
| JUL | 191 | 213 | 250 |
| AUG | 193 | 158 | 326 |
| SEP | 138 | 56 | 116 |
| OCT | 122 | 32 | 44 |

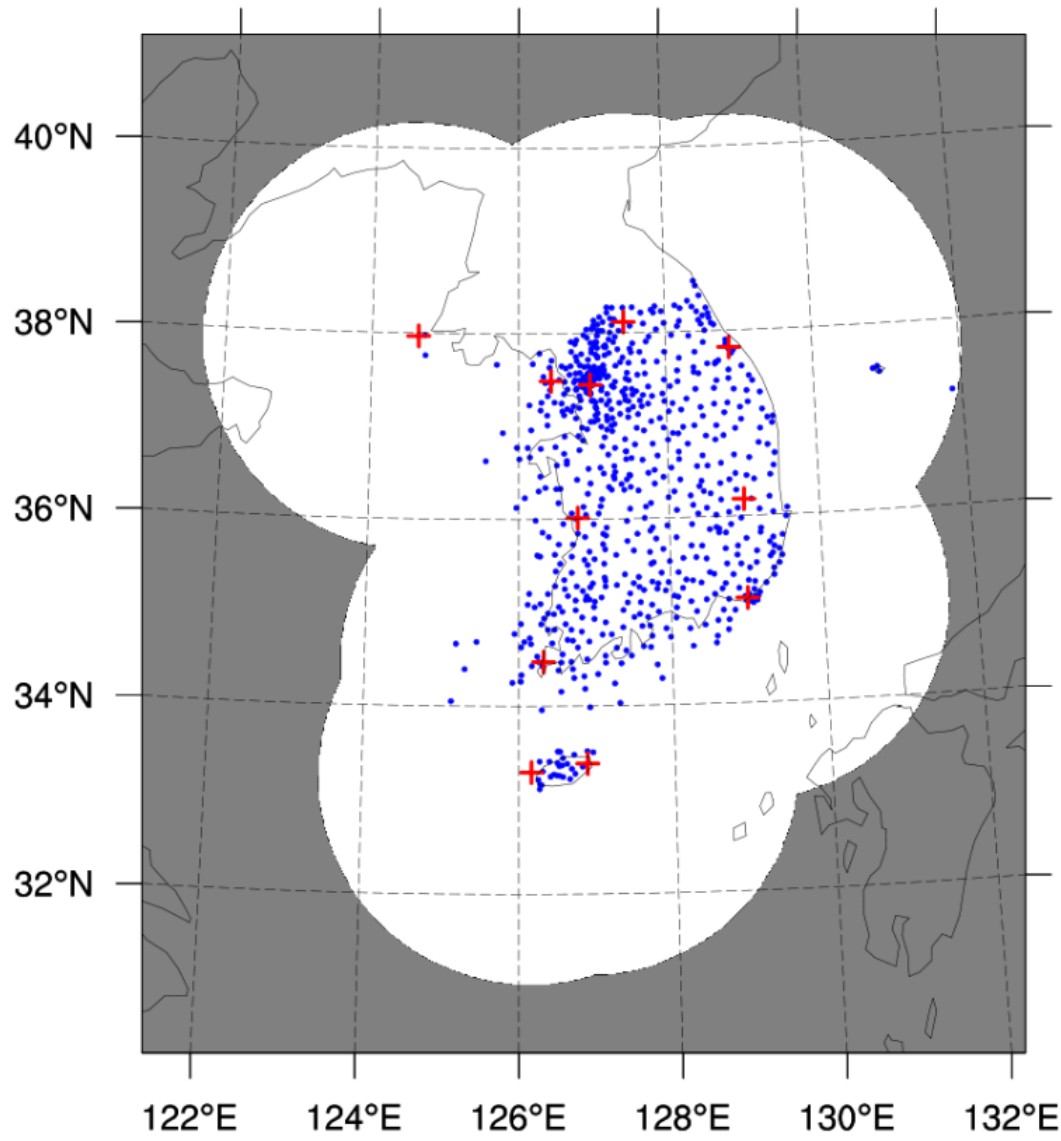

**Figure 1.** The weather station locations (blue dots) and radar locations (red "**+**" symbols) and coverages (white area) in Korea.

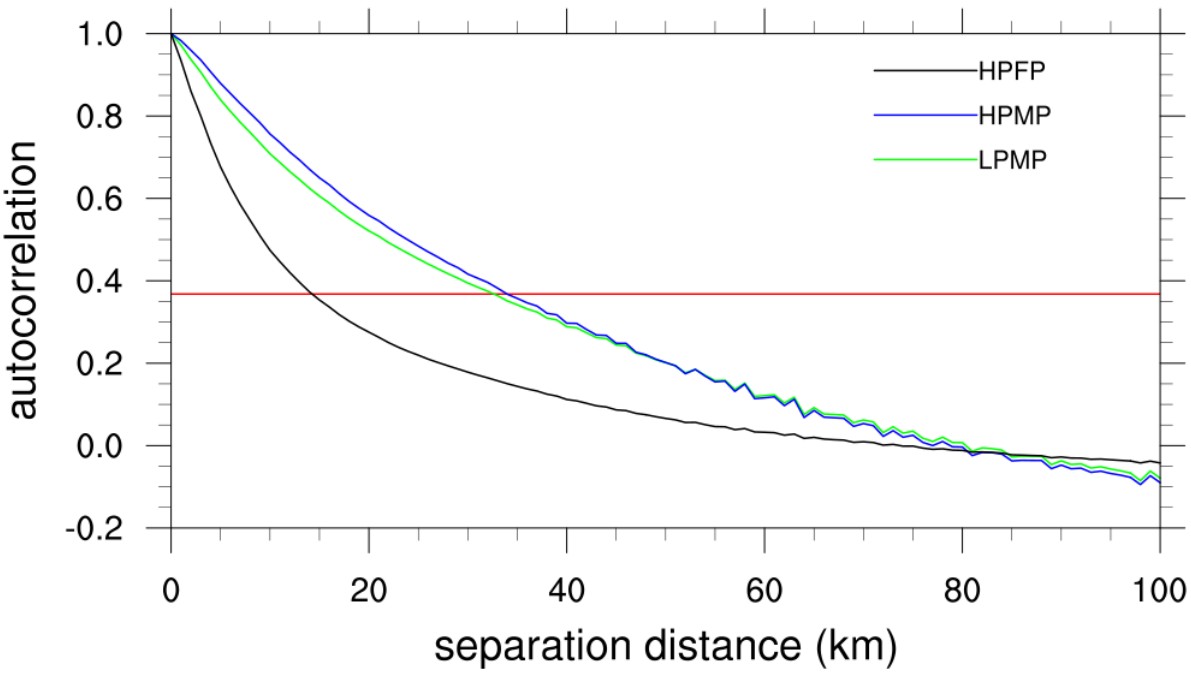

**Figure 2.** The averaged spatial autocorrelations of precipitation for each precipitation type. The red line indicates the $e$-folding value.

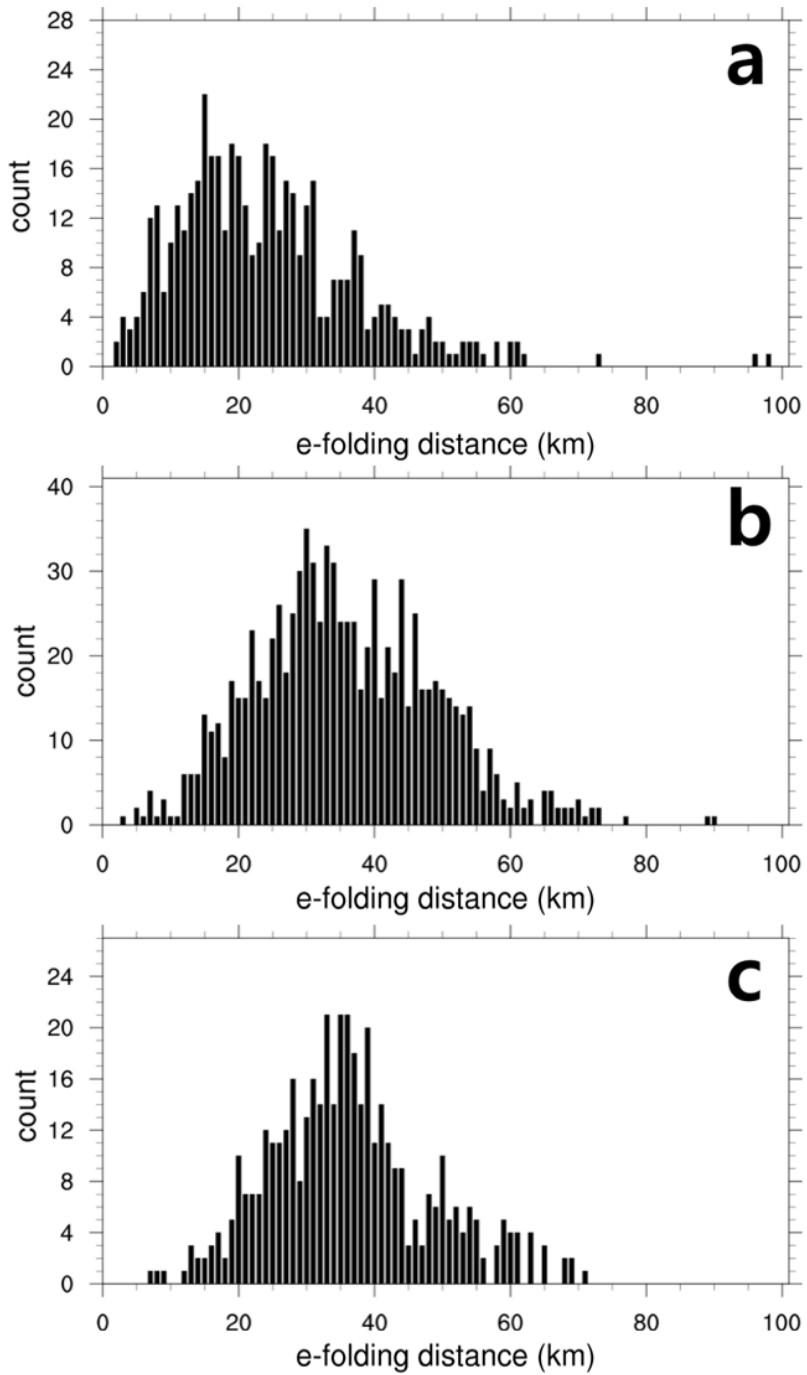

**Figure 3.** A histogram depicting the number of cases according to the *e*-folding distance for (a) HPFP, (b) LPMP, and (c) HPMP.

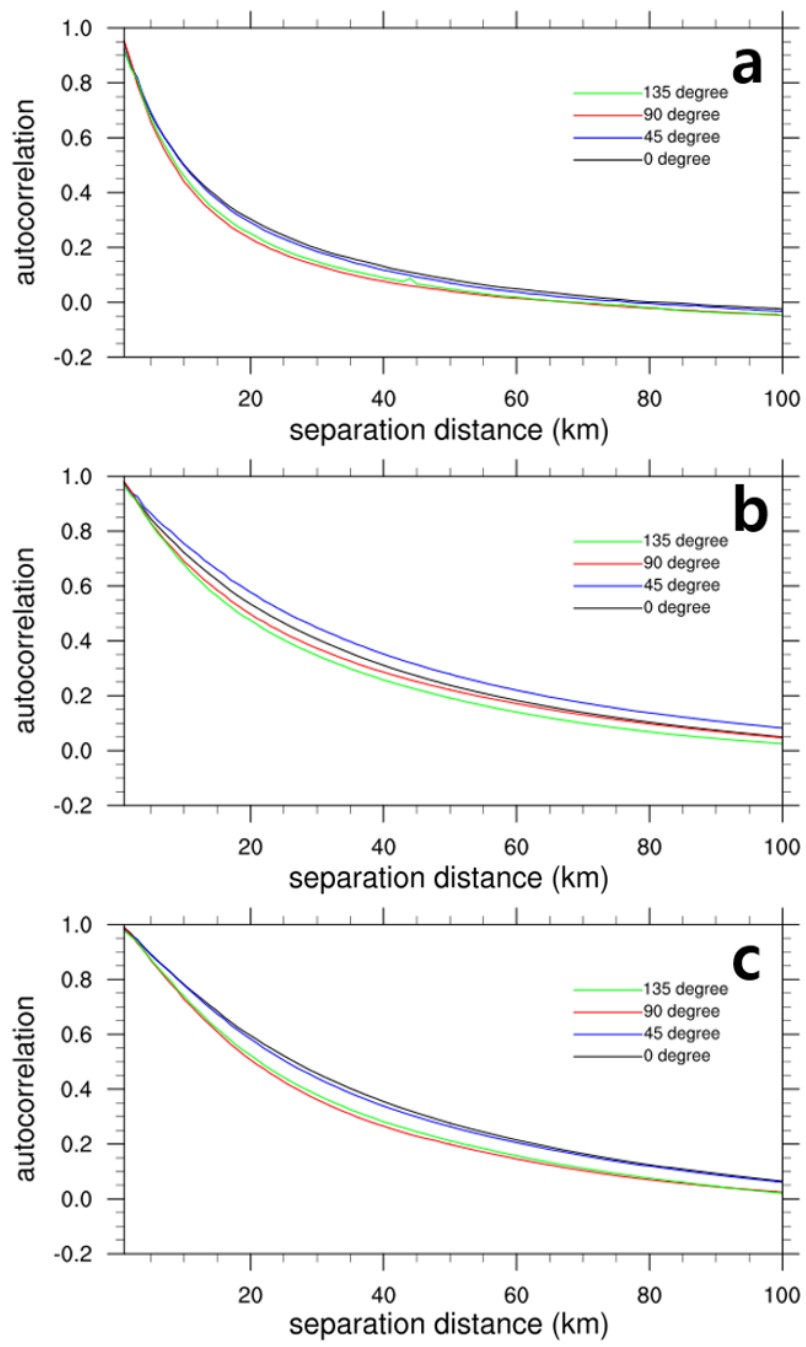

**Figure 4.** The averaged directional spatial autocorrelation of precipitation for (a) HPFP, (b) LPMP, and (c) HPMP.

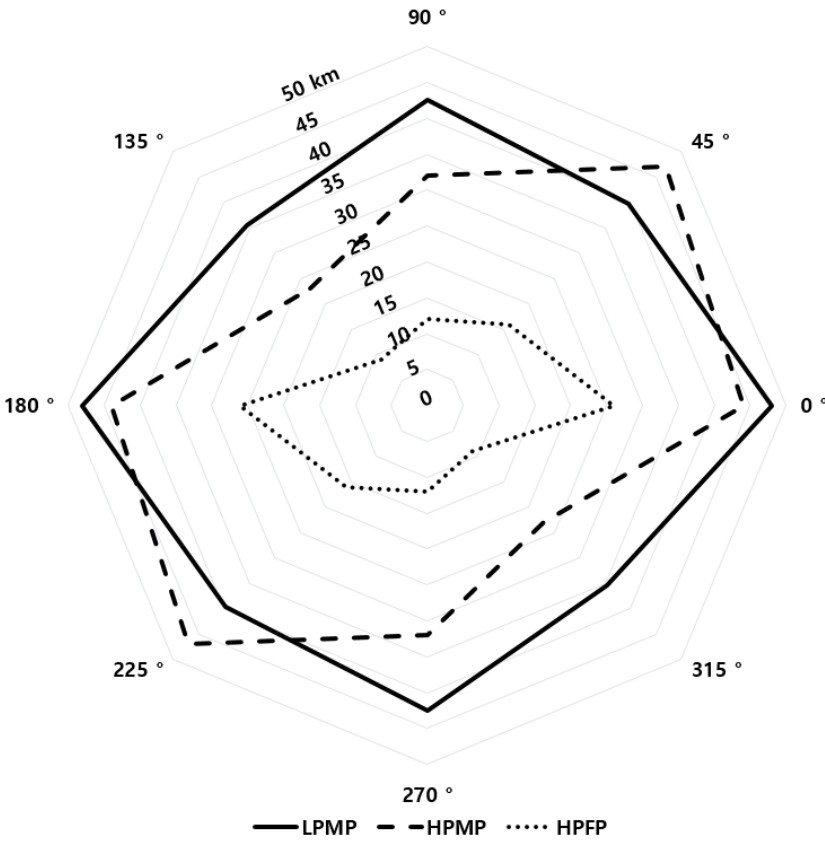

**Figure 5.** A radar chart representing the directional *e*-folding distance (in km) for each precipitation type at the mode in the directional histogram (not shown) depicting the number of cases according to the *e*-folding distance in different directions.

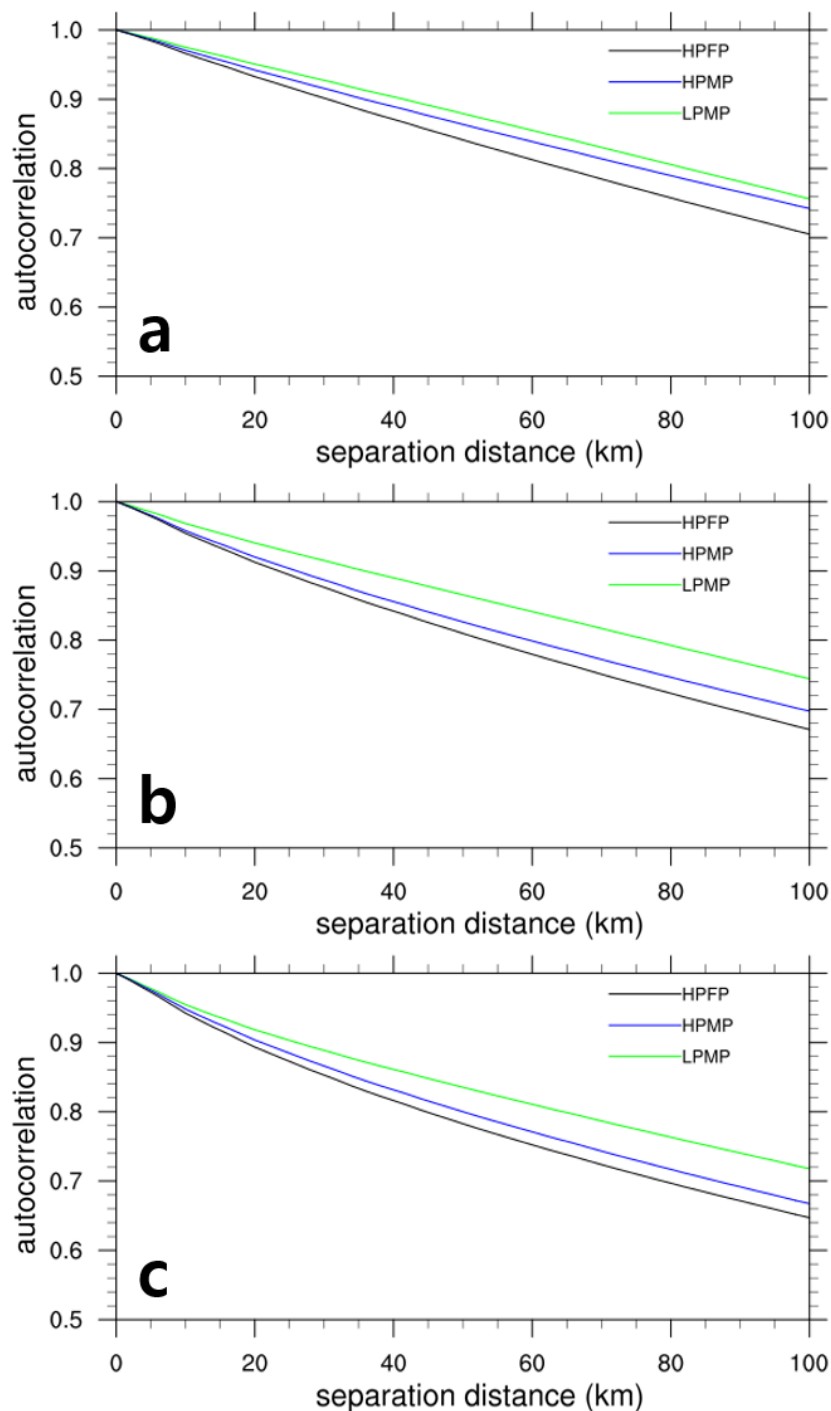

**Figure 6.** The averaged spatial autocorrelation of brightness temperature of the Himawari/AHI water vapor band (a) 8, (b) 9, and (c) 10 for each precipitation type.

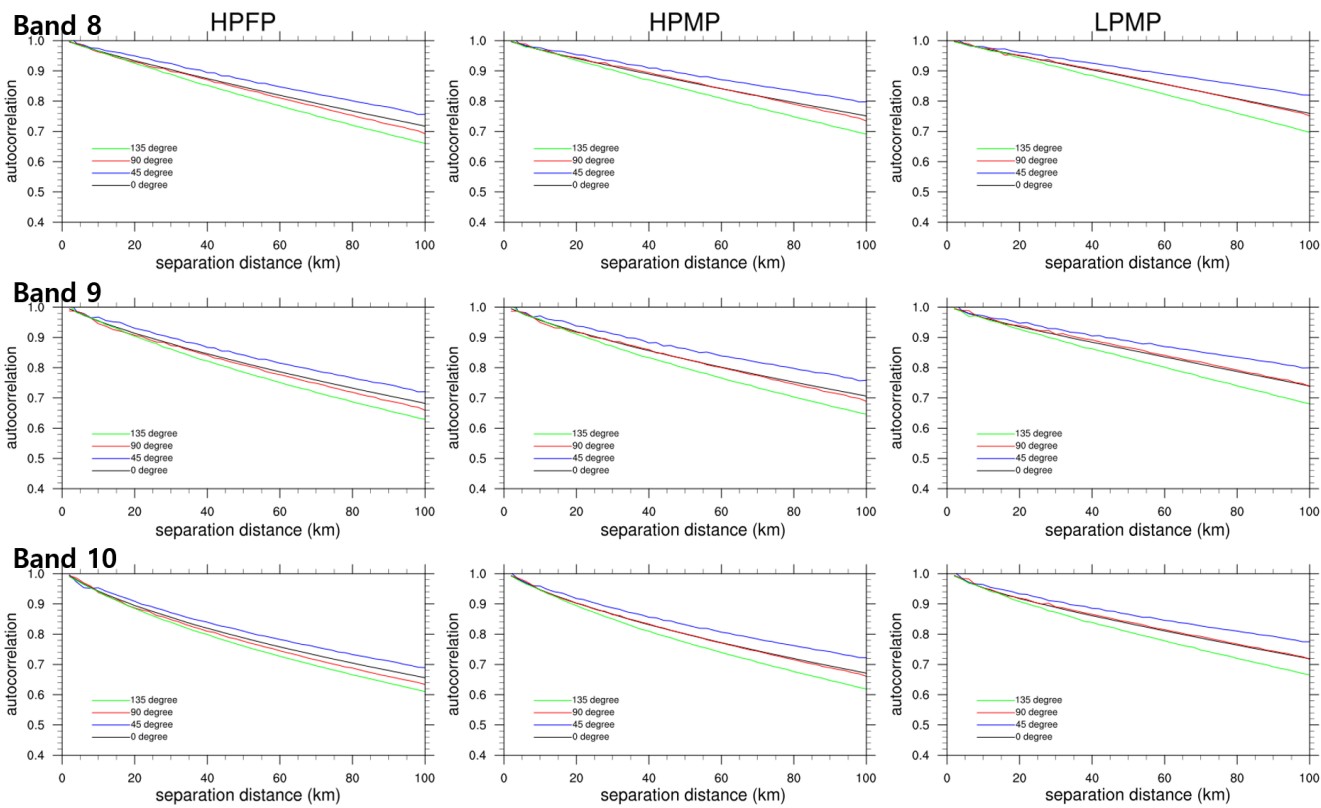

**Figure 7.** The averaged directional spatial autocorrelation of the brightness temperature of the Himawari/AHI water vapor band 8, 9 and 10 by each precipitation type (i.e., HPFP, HPMP, and LPMP) for directions of 0° (black), 45° (blue), 90° (red), and 135° (green). The direction (angle) is measured counterclockwise from the origin–east axis (i.e., 0°).

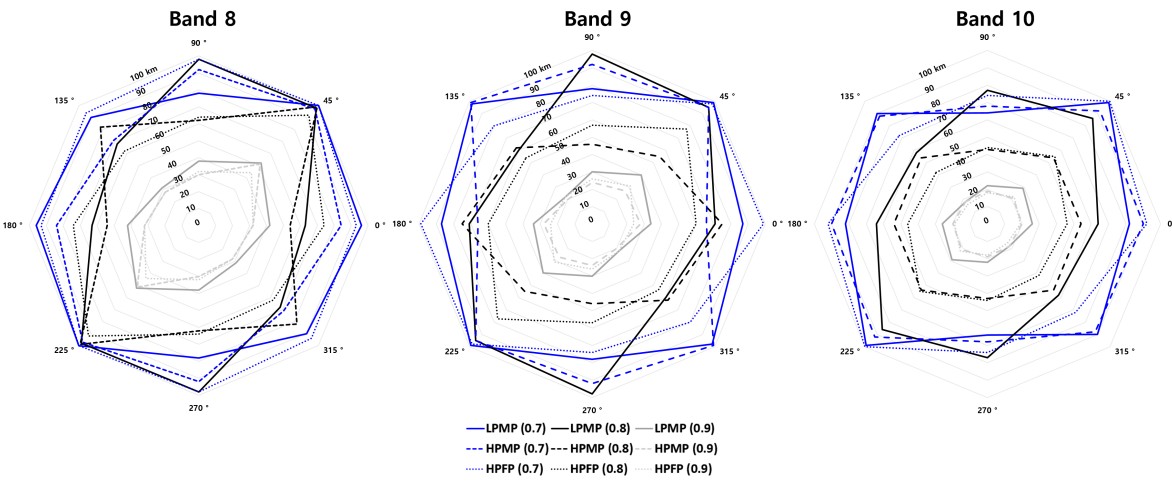

**Figure 8.** A radar chart representing the directional separation distance (in km) of brightness temperature of the Himawari/AHI water vapor band 8, 9, and 10 at the mode in the histogram of case numbers for LPMP (solid), HPMP (dashed), and HPFP (dotted). The colors indicate autocorrelation coefficients of 0.7 (blue), 0.8 (black), and 0.9 (grey), respectively.

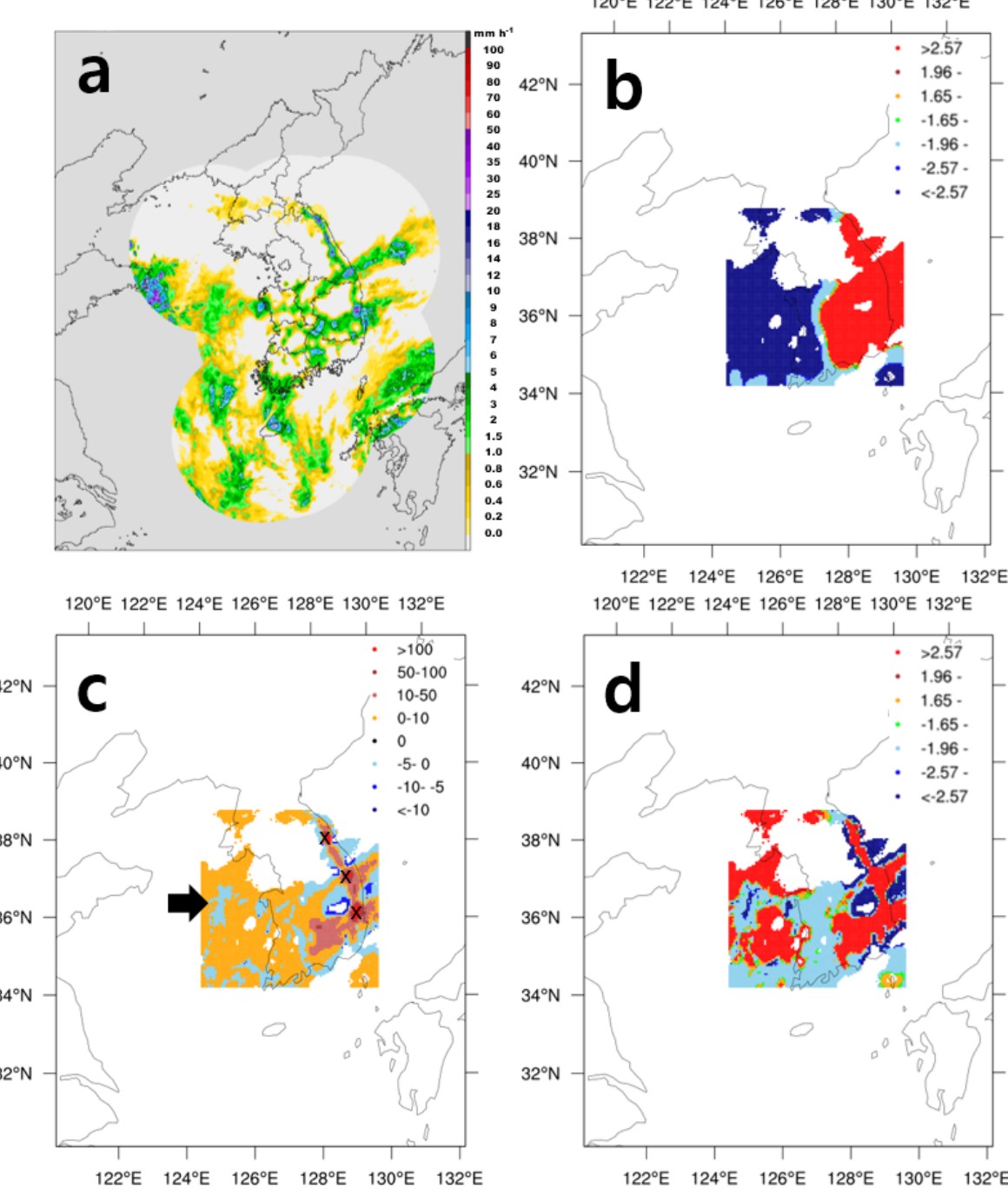

**Figure 9.** An LPMP case at 05 KST 25 August 2015: (a) Precipitation distribution (from https://afso.kma.go.kr/), (b) local $Z(G_i)$, (c) local Moran's $I$ ($I_i$), and (d) Z-score of $I_i$. The computational domain covers the area of 34.34–38.97°N and 124.25–130.05°E. Precipitation systems with maximum intensity and strong cluster characteristics are marked by the X symbols, and the cold spots with dispersion pattern are denoted by the arrow. Non-precipitating areas have no color shading.

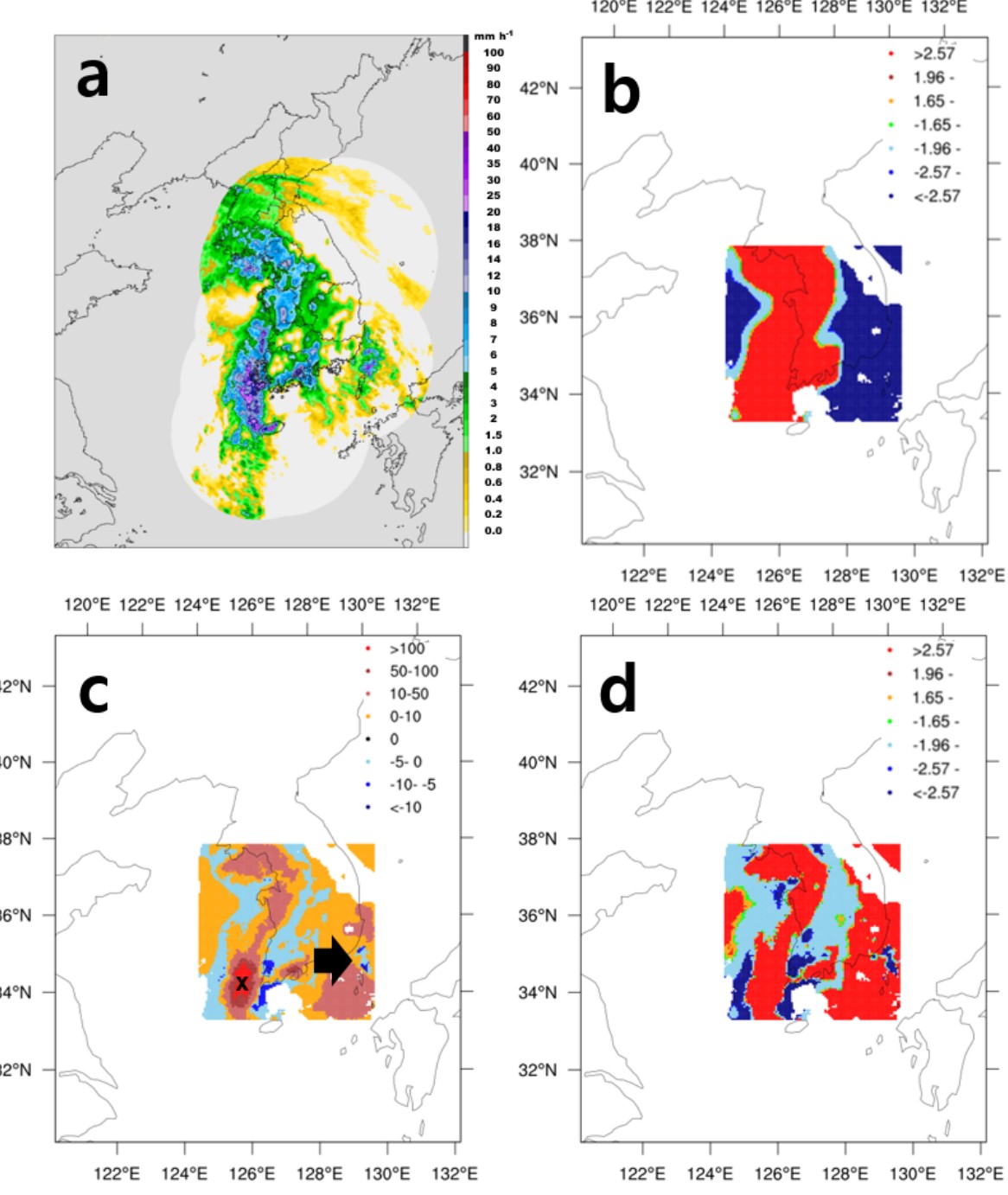

**Figure 10.** Same as in Fig. 9 but for an HPMP case at 17 KST 27 May 2013 and the computational domain of 33.43–38.05°N and 124.25–130.04°E.

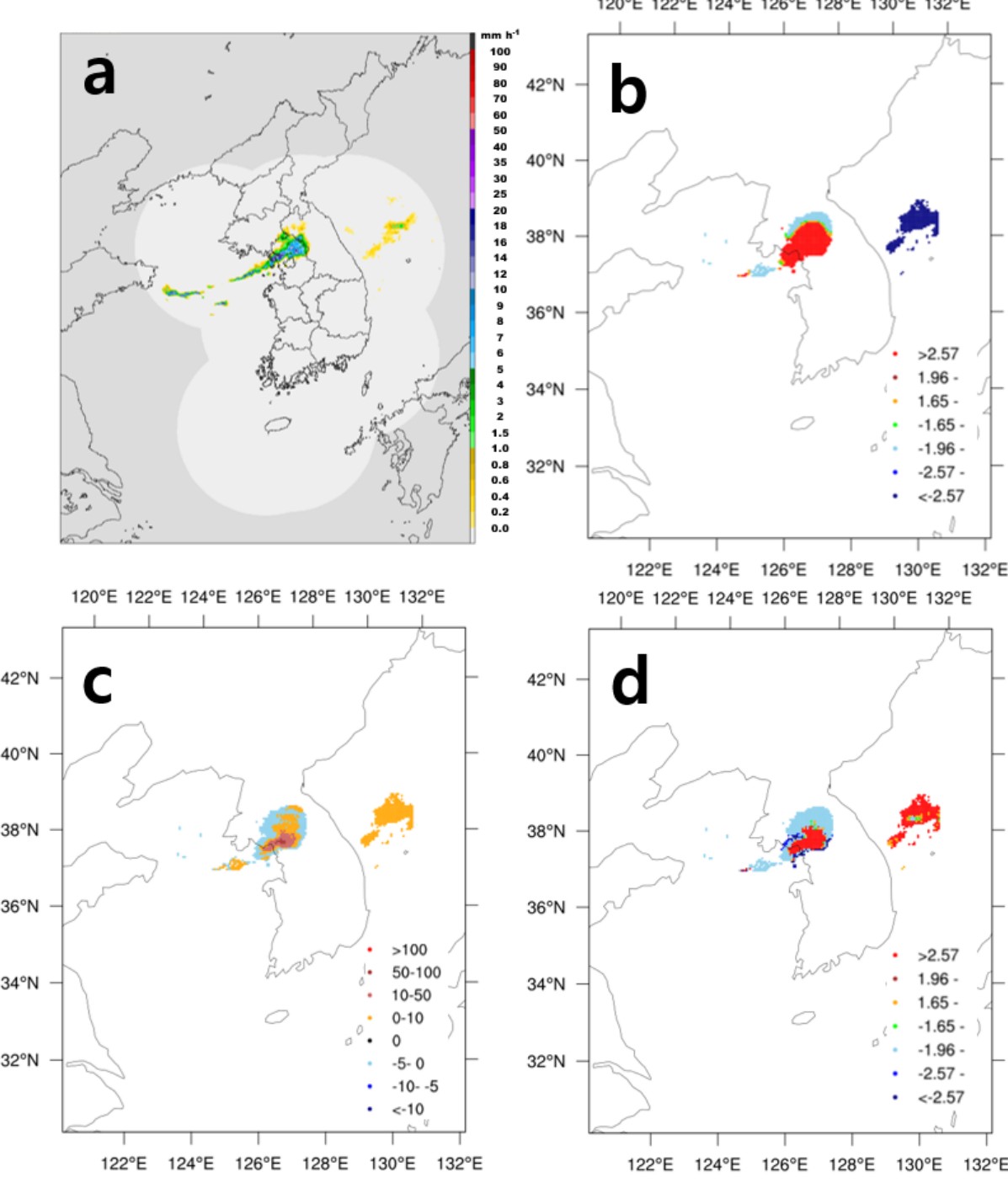

**Figure 11.** Same as in Fig. 9 but for an HPFP case at 05 KST 24 October 2015 and the computational domain of 37.14–39.06°N and 123.32–131.21°E.

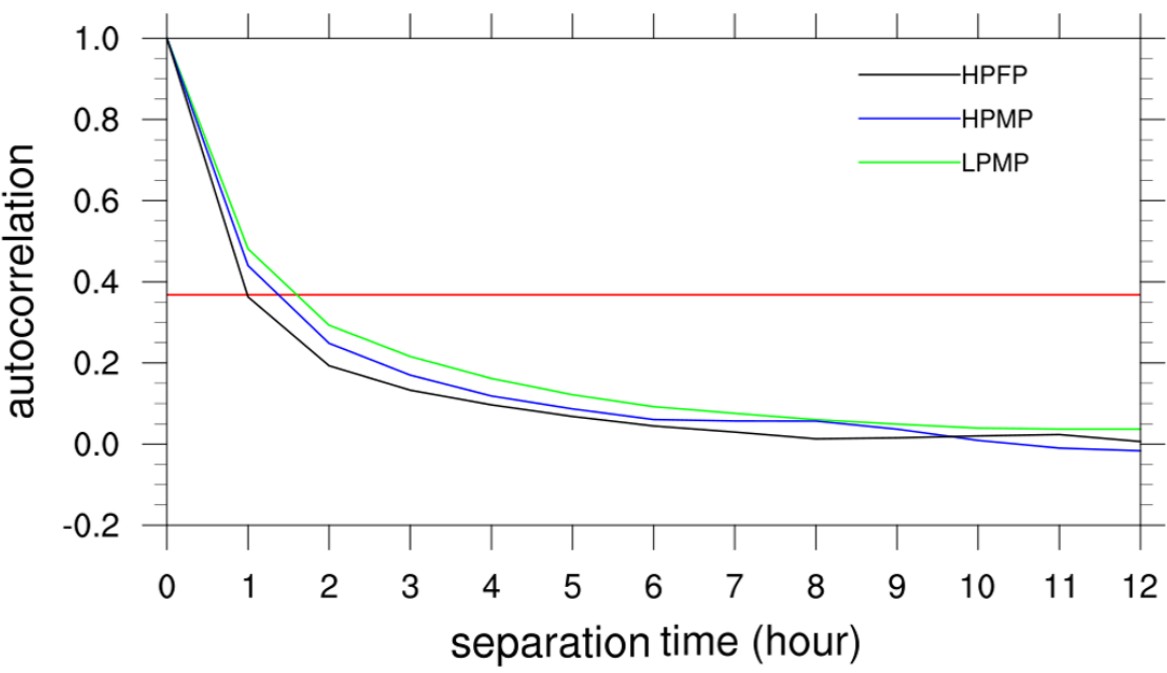

**Figure 12.** Same as in Fig. 2 but for the averaged temporal autocorrelation.

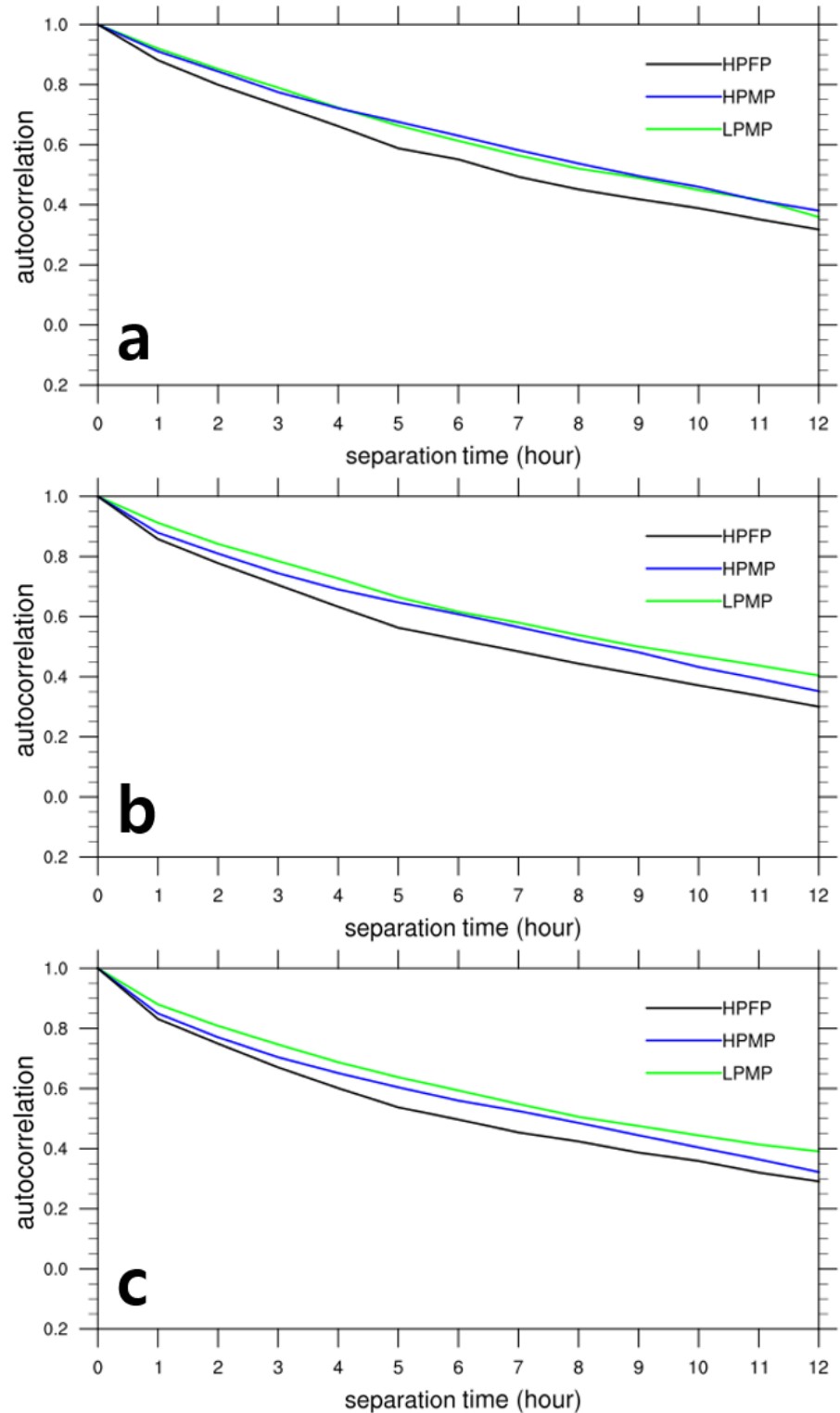

**Figure 13.** Same as in Fig. 6 but for the averaged temporal autocorrelation.

