# Peer review of "Geostatistical assessment of warm-season precipitation observations in Korea based on the composite precipitation and satellite water vapor data"

_Hydrology and Earth System Sciences, 2018_

## Referee Comment (RC1) · X. Yu (Referee) · 28 Mar 2018

General comments:

This manuscript assess the summertime rainfall characters in Korea via the geostatistical analyses on the composite precipitation and satellite water vapor data. Results show that the e-folding distance of precipitation ranges between 15 and 35 km while the e-folding time ranges between 1 and 2 h. The spatial autocorrelation has characteristic directionality. The results show that the current observational network with spation resolution of ~13 km is difficult to capture the characteristic features of the localized heavy rainfall systems. It is also noted that the orographic effect is not assessed in this

manuscript.

The manuscript is well organized and I suggest accept with minor revision.

Specific comments: 1)Page 2, line11-14, this sentence is too long. 2)Page 12, line 9: "Korea" should not be subject. 3)Page 27, in Figure9 caption, a simple description of the case will be better, same for Fig 10-11.

---

## Referee Comment (RC2) · Anonymous Referee #2 · 3 Apr 2018

Reviewer (Comments to Author): This manuscript investigates the spatio-temporal characteristics of summer precipitation systems over the Korean peninsula through the geostatistical analysis using the combined datasets of ground observation and radar data. For the detailed analysis, they categorized the precipitation systems into four types based on the precipitation intensity (3mm/h) and ratio (20%) of precipitated stations. They found that the e-folding distance and time of precipitation systems are clearly dependent on the precipitation area, and directional pattern of precipitation systems. Also they found that the spatial distribution of water vapor has similar characteristics with precipitation but with strong spatial correlation over much longer distance (∼100 km), through the analysis of water vapor channel data of Himawari/Advanced

[Figure]

Himawari Imager data. The results obtained in this study can be used for the detailed understanding of precipitation over South Korea. However, the manuscript should be improved in terms of additional analyses and scientific interpretation of results. Therefore, the manuscript needs to undergo a minor revision before being ready for publication in Hydrology and Earth System Sciences. Below I give some comments and suggestions that would help improving the manuscript.

General comments 1) As we know that thresholds values are very important for the categorization (or clustering) precipitation systems. Please presents the background or ground of threshold values (3mm/h, 20%) used in this study. 2) The domain of data mentioned in the 2 Data description is not well matched with the analysis results (e.g., Figure 1) 3) The author should mention about the sensitivity of analysis results to the threshold values for the categorization of precipitation systems. 4) It will be helpful for the understanding of the single cell storms marked by X in Figures 9 and 10 if the authors presents the background for the marking.

Minor Comments 1) The location of AWS is not correct in Figure 1.

2) I think that the number of y axis in Figures 5 and 8 is km. So, give the unit in Figures 5 and 8.

3) Figure 6. The averaged spatial autocorrelation of brightness temperature of water vapor band (a) 8, (b) 9, and (c) 10 for each precipitation ⇒ The averaged spatial autocorrelation of brightness temperature of water vapor band (a) 8, (b) 9, and (c) 10 Himawari/AHI for each precipitation

4) Some index of figures are not clear (e.g., Figures 7, 8, 9, 10, and 11).

5) Check the order of references (e.g., Cassardo, …, Carruthers, …; Sminth,…Skojen,…)

Please also note the supplement to this comment:

[Figure]

https://www.hydrol-earth-syst-sci-discuss.net/hess-2018-83/hess-2018-83-RC2-supplement.pdf

---

## Referee Comment (RC3) · Anonymous Referee #3 · 10 Apr 2018

This study classifies rainfall systems in Korea on the basis of their spatiotemporal structures by analyzing the observed precipitation, water vapor and cloud data. The subject is of an interest to not only to meteorologists but also to hydrologists. The paper is well organized but needs improvements before it can be accepted for publication. It also needs substantial improvements in writing. Below are my specific comments:

(1) It looks that the rainfall analysis utilizes only the 1-km KMA analysis data. If so, Fig. 1 may not be needed - it can confuse some readers. Please see the comment (2) below for related concerns.

(2) Page 20, line 24: "the portion of weather stations" -> not clear what "weather sta-

[Figure]

tions" imply here. If this implies the 600 real stations in Fig. 1, the authors have to provide how they applied their analysis tools to the irregularly distributed stations. If it implies each grid points of the KMA analysis, this must be indicated ("grid points" instead of 'weather stations") and delete Fig. 1.

(3) Provide the formulation of the weighting function w_ij(d) in Eq. (7) and explain why the specific form is selected to represent the spatial variations.

(4) Page 7, line 20: This result is trivial consequence of classifying the precipitation system in terms of a number of data points; if rainfall occurs only over a small number of points, its spatial scale is limited by design. Their selection of 3 mm/h as the threshold value between heavy and light precipitation may cause the lack of relationship between precipitation intensity and spatial scales. Nam et al. may be a good reference for this. In fact, this result can depend on the selection of the threshold value. The authors need to explain the choice of 3 mm/h as the threshold.

(5) Page 7, line 22 - Page 8, line 15: The authors need to clearly state how the analyses in this block are related to the "propagation of precipitation systems". Analyses in this block are directly related to spatial structures (e.g., shape and orientation); please explain how can these features be related to "propagation".

(6) The spatial shape differences between the three rainfall types (in the same block as above): the asymmetry indicated in the spatial correlation (Fig. 4) does not correspond well to that depicted in the radar echo (Fig. 5). The directional difference in the e-folding scale for the all three systems are about 25% (5km/20km for HPFP; 10km/40km for HPMP and LPMP) 0f the mean scale (i.e., aspect ratios of $\sim$1.3) while the radar echoes suggest larger aspect ratios for HPFP and HPMP ($\sim$1). This is not consistent with their interpretation of rainfall system in Page 8, line 10: how often a squall line is of an aspect ration of 2?

(7) Page 8: If the satellite data cannot clearly distinguish the areas of water vapor from those without, how much can we trust the analysis based on the data? Can they

provide data quality control of the satellite data?

(8) Page 8, line 33: The only similarity between Fig. 2 and Fig. 6 is that the autocorrelation for HPFP decreases more rapidly than those for HPMP and LPMP. The separation between HPFP and HPMP/LPMP in Fig. 6 is much smaller than in Fig. 2 as well. Overall, its difficult to establish similarity between the spatial scales of water vapor and rainfall. The authors need to provide clear explanations on how to related the structures based on water vapor scales (Fig. 6) to that based on rainfall (Fig. 2). Overall, it is difficult to to much merits of the satellite vapor analysis towards the rainfall structure over Korea.

(9) Temporal correlation analysis: It's nor clear what we can learn about the rainfall systems from the temporal correlation characteristics. The e-folding scale differs by only 30 mins among the three types. The time scale of 1-1.5 hours seem to indicate that the all three rainfall types are related to convective systems, either isolated or clustered. Does this provide any insights to separate the characteristics of the three rainfall types? Also, it's not clear how the water vapor analysis can be related to the rainfall characteristics.

(10) Considering the aspect ration and spatial scales, the examples in Fig. 11 seem more relevant for convective clusters (may be imbedded within a frontal structure) than a frontal system.

---

## Author Comment (AC1) · 7 May 2018

**Reply to the Comments by Referee 1 for Manuscript hess-2018-83**

**General comments:** This manuscript assess the summertime rainfall characters in Korea via the geostatistical analyses on the composite precipitation and satellite water vapor data. Results show that the e-folding distance of precipitation ranges between 15 and 35 km while the e-folding time ranges between 1 and 2 h. The spatial autocorrelation has characteristic directionality. The results show that the current observational network with spation resolution of 13 km is difficult to capture the
characteristic features of the localized heavy rainfall systems. It is also noted that the orographic effect is not assessed in this manuscript. The manuscript is well organized and I suggest accept with minor revision.

 $\implies$  We appreciate the positive comments by the referee. We have improved the manuscript by clearly describing some unclear sentences and figure captions. In the following, we made an item-by-item response to the specific comments by the referee.

**Specific comments:**

- 1) Page 2, line 11-14, this sentence is too long.
  - $\implies$  We have rewritten this part as:

Capturing the spatiotemporal features of precipitation systems out of the observation networks is essential to the successful runoff forecast, especially at the catchment scale and for the flooding cases (Volkmann et al., 2010).

- 2) Page 12, line 9: "Korea" should not be subject.
  - $\implies$  We have changed this part to:

Currently, the precipitation observation network in Korea has a spatial resolution of  ${\sim}13$  km, distributing the analysis in an interval of 1 h.

- 3) Page 27, in Figure 9 caption, a simple description of the case will be better, same for Fig 10–11.
  - $\implies$  We have given a description of the case by including the precipitation type in each figure; however, we kept the information on time and date for each
event because it is usually requested by the readers for the case studies. We have rewritten the figure's by reflecting other referee's comments as well. The modified captions, with new statements in bold, are as the followings:

Figure 9. An LPMP case at 05 KST 25 August 2015: (a) Precipitation distribution (source from https://afso.kma.go.kr/), (b) local  $Z(G_i)$ , (c) local Moran's  $I(I_i)$ , and (d) Z-score of  $I_i$ . The computational domain covers the area of  $34.34 - 38.97^{\circ}$  N and  $124.25 - 130.05^{\circ}$  E. Precipitation systems with maximum intensity and strong cluster characteristics are marked by the crosses, and the cold spots with dispersion pattern are denoted by the arrow. Non-precipitating areas have no color shading.

Figure 10. Same as in Fig. 9 but for an HPMP case at 17 KST 27 May 2013 and the computational domain of  $33.43 - 38.05^{\circ}$ N and  $124.25 - 130.04^{\circ}$ E.

Figure 11. Same as in Fig. 9 but for an HPFP case at 05 KST 24 October 2015 and the computational domain of  $37.14 - 39.06^{\circ}$ N and  $123.32 - 131.21^{\circ}$ E.

HESSD

---

## Author Comment (AC2) · 7 May 2018

**Reply to the Comments by Referee #2 for Manuscript hess-2018-83**

*Reviewer (Comments to Author):*
*This manuscript investigates the spatio-temporal characteristics of summer precipitation systems over the Korean peninsula through the geostatistical analysis using the combined datasets of ground observation and radar data. For the detailed analysis, they categorized the precipitation systems into four types based on the precipitation intensity (3mm/h) and ratio (20%) of precipitated stations. They found that the e-folding*

*distance and time of precipitation systems are clearly dependent on the precipitation area, and directional pattern of precipitation systems. Also they found that the spatial distribution of water vapor has similar characteristics with precipitation but with strong spatial correlation over much longer distance ($\sim$100 km), through the analysis of water vapor channel data of Himawari/Advanced Himawari Imager data. The results obtained in this study can be used for the detailed understanding of precipitation over South Korea. However, the manuscript should be improved in terms of additional analyses and scientific interpretation of results. Therefore, the manuscript needs to undergo a minor revision before being ready for publication in Hydrology and Earth System Sciences. Below I give some comments and suggestions that would help improving the manuscript.*

$\Longrightarrow$ We appreciate the positive and valuable comments by the referee. We have substantially improved the manuscript by making some unclear statements clearer and by adding more discussions on detailed analyses and scientific interpretation, following the referee's comments. An item-by-item response to the referee's general and minor comments is provided below.

**General comments:**

1) *As we know that thresholds values are very important for the categorization (or clustering) precipitation systems. Please presents the background or ground of threshold values (3mm/h, 20%) used in this study.*

   $\Longrightarrow$ We agree with the referee that the threshold values are important for categorizing the precipitation systems. In order to determine the threshold values (i.e., 20% and 3 mm/h), we have performed a preliminary statistical analysis of precipitation events (see Table R1 below). In classifying the precipitation types, we used two criteria — the portion of weather stations with precipita-

tion (C1) and the station average precipitation rate (C2). We determined the
threshold values when the cumulative percentage of each criterion reaches
80% (see the red lines in Table R1). For example, in terms of C1, the cumu-
lative percentage reaches 77.1% with the portion of 10–20% and 85.0% with
the portion of 20–30%; thus selecting 20% as the threshold value. In terms
of C2, the cumulative percentage becomes 80.0% with 2.0–2.9 mm/h and
93.3% with 3.0–4.9 mm/h; thus choosing 3 mm/h as the threshold value. We
have added the following statement at the early part of Sec. 2 in the revised
manuscript.

> In order to determine the threshold values for classifying the pre-
> cipitation types, we have conducted a preliminary statistical analy-
> sis on precipitation events in the period of 2011–2015 (not shown).
> As the precipitation events occur in a given time period and/or
> space interval, our precipitation data are assumed to follow the
> Poisson distribution, which represents a probability situation of a
> large number of observation with a small probability of occurrence.
> Many studies developed the Poisson distribution models to estimate
> rainfall and cluster the rainfall systems (e.g., Rodriguez-Iturbe et al.,
> 1987; Lee et al., 2014; Barton et al., 2016; Ritschel et al., 2017). We
> have chosen the threshold values when the cumulative percentage
> of precipitation events for each criterion (i.e., C1 and C2) reached
> approximately 80%; thus obtaining the threshold values of 20% for
> C1 and 3 mm h$^{-1}$ for C2, respectively.

2) *The domain of data mentioned in the 2 Data description is not well matched with
   the analysis results (e.g., Figure 1)*

⟹ We appreciate the referee for pointing this out, and we admit that our data
description and Fig. 1 might have caused confusion. We have used the sta-
tion data, as shown in Fig. 1, to classify the precipitation types (see Table

R1 below); we have utilized the 1 km composite precipitation data for the precipitation analyses, including spatial correlations. We actually noticed that Fig. 1 should be updated because the station precipitation data were obtained from three observation networks with a total of 688 stations — the Automated Synoptic Observing Systems (ASOS), the Automatic Weather Stations (AWS), and Automated Agriculture Observing System (AAOS). We also noticed that the information on the radar locations and coverages would be essential because both the station and radar data were used to produce the 1 km composite precipitation data. In the revised manuscript, we modified Fig. 1 by updating the weather station locations and by including the radar locations and coverages (see Fig. R1 below). We have rewritten the text by clearly describing the data used in this study. We first modified the beginning sentences in the second paragraph of Sec. 1, with new statements in bold, as:

> "The ground-based rainfall observation data, in Korea, are collected from the Automated Synoptic Observing Systems (ASOS), the Automatic Weather Stations (AWS), **and the Automated Agriculture Observing System (AAOS)**. The observation density is about 67 km for ASOS and approximately 13 km by including AWS. **In addition, the agrometeorological observation network consists of 11 AAOS stations (Choi et al., 2015).** · · ·"

⟹ We have also modified and reorganized the early part of Sec. 2, by including the step-by-step description of the method to produce the composite precipitation data, as (new sentences in bold):

> **We used the precipitation data from weather stations, shown in Fig. 1, to categorize the precipitation systems.** We classify four different precipitation types statistically based on two criteria: the portion of weather stations with precipitation (C1), and the station average precipitation rate (C2). Based on these criteria,

we define four different precipitation types, as shown in Table 1: 1) Low Precipitation at a Few Points (LPFP) for C1 < 20 % and C2 < 3 mm h$^{-1}$; 2) Low Precipitation at Many Points (LPMP) for C1 ≥ 20 % and C2 < 3 mm h$^{-1}$; 3) High Precipitation at a Few Points (HPFP) for C1 < 20 % and C2 ≥ 3 mm h$^{-1}$; and 4) High Precipitation at Many Points (HPMP) for C1 ≥ 20 % and C2 ≥ 3 mm h$^{-1}$. We practically exclude the LPFP type in our analyses, i.e., the case with C1 < 20 % and C2 < 3 mm h$^{-1}$, because it may be less effective.

The Korea Meteorological Administration (KMA) has produced a composite precipitation data over Korea **using the data from radars, weather stations and satellites, through the following steps as described in Hwang et al. (2015): 1) remove non-precipitation echoes from the radar data using the satellite cloud type data; 2) calculate the difference between the station precipitation and the radar estimated precipitation; 3) perform the objective analysis on the precipitation difference field and on the station precipitation data; 4) correct the bias using the objectively-analyzed difference field; and 5) combine the corrected radar-estimated precipitation data and the objectively-analyzed station precipitation data to produce the composite precipitation data (in mm h$^{-1}$). In order to analyze the precipitation systems with high resolution and evenly distributed data, we used this composite precipitation data.** This data covers 1153 km × 1441 km over the Korean Peninsula, with a grid size of 1 km and a time resolution of 1 h. Geostatistical analyses are conducted using this composite precipitation data sets from April to October in a period of 2013–2015 to investigate the spatial and temporal characteristics of summer rainfall.

3) *The author should mention about the sensitivity of analysis results to the threshold values for the categorization of precipitation systems.*

⟹ As shown in Table R1, heavy precipitation systems have high locality; especially, precipitation with the highest intensity ($\geq 10$ mm/hr) mostly occurs in a small area with the number of stations less than 10% of total weather stations. This is consistent with the findings of Nam et al. (2014), and implies that the precipitation analysis results may depend on (be sensitive to) the threshold values. We have added the following sentence to next to the newly-added paragraph in item 1) above:

> Our preliminary statistical analysis showed that, in general, most precipitation events occur over small areas and precipitation events with high intensity rarely occur over large areas. The locality of precipitation appeared higher as the precipitation intensity were higher, in accordance with Nam et al. (2014). In particular, precipitation systems with the highest intensity ($\geq 10$ mm/hr) were mostly confined to a small area with the number of stations less than 10% of total weather stations. This implies that the locality feature of precipitation systems may depend on the threshold value in precipitation intensity.

4) *It will be helpful for the understanding of the single cell storms marked by X in Figures 9 and 10 if the authors presents the background for the marking.*

⟹ We appreciate the referee for pointing this out. We think the expression "single cell storms" is not appropriate here. We originally intended to put the "X" marks on the locations of precipitation systems with maximum intensity (precipitation rate) and strong cluster characteristics. To avoid any confusion, we have modified the captions of Figs. 9 and 10 accordingly. By reflecting the referee's suggestion in Minor comments (items 3 and 4) and

other referee's comments as well, the captions are rewritten as:

> Figure 9. **An LPMP case at 05 KST 25 August 2015:** (a) Precipitation distribution (source from https://afso.kma.go.kr/), (b) local Z($G_i$), (c) local Moran's $I$ ($I_i$), and (d) Z-score of $I_i$. **The computational domain covers the area of** $34.34 - 38.97°$ **N and** $124.25 - 130.05°$ **E. Precipitation systems with maximum intensity and strong cluster characteristics** are marked by the crosses, and the cold spots with dispersion pattern are denoted by the arrow. **Non-precipitating areas have no color shading.**
>
> Figure 10. Same as in Fig. 9 but **for an HPMP case at 17 KST 27 May 2013 and the computational domain of** $33.43 - 38.05°$**N and** $124.25 - 130.04°$**E**.

$\implies$ We have also rewritten the statement in page 10, line 11 as:

> Figure 9c depicts **several precipitation systems with maximum intensity (i.e., precipitation rate; see Fig. 9a)** in the cluster area (marked by the crosses). **These systems show the highest local Moran's $I$ with the spatial scale of less than 30 km.** The cluster patterns were statistically significant at a significance level of 0.01 (Fig. 9d). $\cdots\cdots$ The map of local Moran's $I$ shows that **a precipitation system with a strong cluster feature has developed**, over the southwestern sea of the Korean Peninsula, along with the successive cluster patterns **in a line type** (Fig. 10c), with a significance level of 0.01 (Fig. 10d).

***Minor comments:***

1) *The location of AWS is not correct in Figure 1.*

⟹ We have redrawn Fig. 1 in the revised manuscript (see Fig. R1 below).

2) *I think that the number of y axis in Figures 5 and 8 is km. So, give the unit in Figures 5 and 8.*

⟹ We appreciate the referee for pointing this out. We now have explicitly given the unit (km) in the captions of Figs. 5 and 8. We also changed the name of diagram to "radar chart" from "radar diagram" to avoid any confusion. The captions of Figs. 5 and 8 are rewritten, and the cation of Fig. 8 also reflects the referee's suggestion in items 3) and 4):

Figure 5. A radar **chart** representing the directional $e$-folding distance **(in km)** at the mode in Fig. 3.

Figure 8. A radar **chart** representing the directional $e$-folding distance **(in km)** of brightness temperature of **the Himawari/AHI** water vapor band **8, 9 and 10** at the mode in the histogram of case numbers for **LPMP (solid), HPMP (dashed), and HPFP (dotted). The colors indicate autocorrelation coefficients of 0.7 (blue), 0.8 (black), and 0.9 (grey), respectively**.

3) *Figure 6. The averaged spatial autocorrelation of brightness temperature of water vapor band (a) 8, (b) 9, and (c) 10 for each precipitation ⟹ The averaged spatial autocorrelation of brightness temperature of water vapor band (a) 8, (b) 9, and (c) 10 Himawari/AHI for each precipitation*

⟹ The caption of Fig. 6 is now modified following the referee's suggestion as:

Figure 6. The averaged spatial autocorrelation of brightness temperature of **the Himawari/AHI** water vapor band (a) 8, (b) 9, and (c) 10 for each precipitation type.

4) *Some index of figures are not clear (e.g., Figures 7, 8, 9, 10, and 11).*

$\Longrightarrow$ We have rewritten the captions, by reflecting suggestions by other referees as well. The revised captions appear as the followings:

Figure 7. The average directional spatial autocorrelation of the brightness temperature of **the Himawari/AHI** water vapor band **8, 9 and 10** by each precipitation type **(i.e., HPFP, HPMP, and LPMP) for directions of** $0°$ **(black),** $45°$ **(blue),** $90°$ **(red), and** $135°$ **(green). The direction (angle) is measured counterclockwise from the origin-east axis (i.e.,** $0°$**).**

Figure 8. A radar **chart** representing the directional $e$-folding distance **(in km)** of brightness temperature of **the Himawari/AHI** water vapor band **8, 9 and 10** at the mode in the histogram of case numbers for **LPMP (solid), HPMP (dashed), and HPFP (dotted). The colors indicate autocorrelation coefficients of 0.7 (blue), 0.8 (black), and 0.9 (grey), respectively**.

Figure 9. **An LPMP case at 05 KST 25 August 2015:** (a) Precipitation distribution (source from https://afso.kma.go.kr/), (b) local Z($G_i$), (c) local Moran's $I$ ($I_i$), and (d) Z-score of $I_i$. **The computational domain covers the area of** $34.34 − 38.97°$ **N and** $124.25 − 130.05°$ **E. Precipitation systems with maximum intensity and strong cluster characteristics** are marked by the crosses, and the cold spots with dispersion pattern are denoted by the arrow. **Non-precipitating areas have no color shading.**

Figure 10. Same as in Fig. 9 but **for an HPMP case at 17 KST 27 May 2013 and the computational domain of** $33.43 − 38.05°$**N and** $124.25 − 130.04°$**E.**

Figure 11. Same as in Fig. 9 but **for an HPFP case at 05 KST 24 October 2015 and the computational domain of** $37.14 - 39.06°$**N and** $123.32 - 131.21°$**E**.

[Figure]

**Table R1.** Preliminary statistical analysis of precipitation events during 2011–2015 by two criteria —- the portion of weather stations with precipitation and the station average precipitation rate. The red lines indicate the boundaries when the cumulative percentage of precipitation events is approximately 80 %.

**Figure R1.** The weather station locations (blue dots) and radar locations (red plus symbols) and coverages (white area) in Korea.

**References**

Barton, Y., Giannakaki, P., Von Waldow, H., Chevalier, C., Pfahl, S., and Martius, O.: Clustering of regional-scale extreme precipitation events in Southern Switzerland, Mon. Wea. Rev., 144, 347–369, 2016.

Choi, S.-W., Lee, S.-J., Kim, J. Lee, B.-L., Kim, K.-R., and Choi, B.-C.: Agrometeorological observation environment and periodic report of Korea Meteorological Administration: Current status and suggestions, Korean J. Agric. For. Meteorol., 17, 144–155, doi:10.5532/KJAFM.2015.17.2.144, 2015 (in Korean with English abstract).

Lee, J., Yoon, J., and Jun, H. D.: Evaluation for the correction of radar rainfall due to the spatial distribution of raingauge network, J. Korea Soc. Hazzard Mitig., 14, 337–345, http://dx.doi.org/10.9798/KOSHAM.2014.14.6.337, 2014.

Nam, J.-E., Lee, Y. H., Ha, J.-C., and Cho, C.-H.: A study on the e-folding distance of summer precipitation using precipitation reanalysis data, in: Proceedings of the Autumn Meeting of Korean Meteorological Society, 2014, Korean Meteorol. Soc., Jeju, Korea, 13–15 October 2014, 657–658, 2014.

Ritschel, C., Ulbrich, U., Névir, P., and Rust, H. W.: Precipitation extremes on multiple timescales — Bartlett-Lewis rectangular pulse model and intensity-duration-frequency curves, Hydrol. Earth Syst. Sci., 21, 6501–6517, https://doi.org/10.5194/hess-21-6501-2017, 2017.

Rodriguez-Iturbe, I., Cox, D. R., and Isham, V.: Some models for rainfall based on stochastic point processes, Proc. R. Soc. London, Ser. A, 410, 269–288, 1987.

| | | The portion of weather stations with precipitation (%) | | | | | | | | | | Sum | Percentage (%) | Cumulative percentage (%) |
|---|---|---|---|---|---|---|---|---|---|---|---|---|---|---|
| | | 0-10 | 10-20 | 20-30 | 30-40 | 40-50 | 50-60 | 60-70 | 70-80 | 80-90 | 90- | | | |
| The station average precipitation rate (mm/h) | 0.1-0.9 | 5241 | 399 | 156 | 60 | 29 | 5 | 2 | 1 | 0 | 0 | 5893 | 40.0 | 40.0 |
| | 1.0-1.9 | 2238 | 665 | 383 | 235 | 148 | 99 | 57 | 33 | 18 | 6 | 3882 | 26.3 | 66.3 |
| | 2.0-2.9 | 891 | 352 | 262 | 196 | 109 | 89 | 48 | 35 | 29 | 12 | 2023 | 13.7 | 80.0 |
| | 3.0-4.9 | 765 | 322 | 224 | 212 | 159 | 85 | 77 | 43 | 49 | 35 | 1971 | 13.4 | 93.3 |
| | 5.0-9.9 | 317 | 118 | 116 | 115 | 76 | 47 | 35 | 19 | 18 | 8 | 869 | 5.9 | 99.2 |
| | 10.0- | 61 | 6 | 19 | 11 | 14 | 1 | 0 | 0 | 0 | 0 | 112 | 0.8 | 100.0 |
| | Sum | 9513 | 1862 | 1160 | 829 | 535 | 326 | 219 | 131 | 114 | 61 | 14750 | | |
| | Percentage (%) | 64.5 | 12.6 | 7.9 | 5.6 | 3.6 | 2.2 | 1.5 | 0.9 | 0.8 | 0.4 | | | |
| | Cumulative percentage (%) | 64.5 | 77.1 | 85.0 | 90.6 | 94.2 | 96.4 | 97.9 | 98.8 | 99.6 | 100.0 | | | |

**Fig. 1.** Table R1. See the caption for Table R1 in C11.

**Fig. 2.** Figure R1. See the caption for Figure R1 in C11.

---

## Author Comment (AC3) · 7 May 2018

**Reply to the Comments by Referee #3 for Manuscript hess-2018-83**

*This study classifies rainfall systems in Korea on the basis of their spatiotemporal structures by analyzing the observed precipitation, water vapor and cloud data. The subject is of an interest to not only to meteorologists but also to hydrologists. The paper is well organized but needs improvements before it can be accepted for publication. It also needs substantial improvements in writing.*

[Figure]

$\implies$ We appreciate the positive and valuable comments by the referee, which helped us improve the quality of the manuscript. We have faithfully revised the manuscript following the referee's specific comments, including some corrections and suggestions. We have also rewritten many parts of the manuscript, trying to avoid any confusion, especially in description of data and interpretation of results. In the following, we made an item-by-item response to the specific comments by the referee.

**Below are my specific comments:**

(1) *It looks that the rainfall analysis utilizes only the 1-km KMA analysis data. If so, Fig. 1 may not be needed - it can confuse some readers. Please see the comment (2) below for related concerns.*

$\implies$ As the referee pointed out, for our rainfall analyses, we have utilized the 1 km composite precipitation data, which were based on both the station and radar data. However, in classifying the precipitation types (see Table R1 below), we have used the station data only, and included Fig. 1. We actually noticed that Fig. 1 should be updated because the station precipitation data included the data from three observation networks with a total of 688 stations — the Automated Synoptic Observing Systems (ASOS), the Automatic Weather Stations (AWS), and Automated Agriculture Observing System (AAOS). We also noticed that the information on the radar locations and coverages would be essential because both the station and radar data were used to produce the 1 km composite precipitation data. In the revised manuscript, we modified Fig. 1 by updating the weather station locations and by including the radar locations and coverages (see Fig. R1 below). We have rewritten the text by clearly describing the data used in this study. We have modified the beginning sentences in the second paragraph of Sec. 1, with new statements in bold, as:

"The ground-based rainfall observation data, in Korea, are collected from the Automated Synoptic Observing Systems (ASOS), the Automatic Weather Stations (AWS), **and the Automated Agriculture Observing System (AAOS)**. The observation density is about 67 km for ASOS and approximately 13 km by including AWS. **In addition, the agrometeorological observation network consists of 11 AAOS stations (Choi et al., 2015).** ···"

See also the early part of Sec. 2, and the authors' reply to the referee's comments (2) below.

(2) *Page 20, line 24: "the portion of weather stations" → not clear what "weather stations" imply here. If this implies the 600 real stations in Fig. 1, the authors have to provide how they applied their analysis tools to the irregularly distributed stations. If it implies each grid points of the KMA analysis, this must be indicated ("grid points" instead of "weather stations") and delete Fig. 1.*

$\implies$ We assume that the referee meant for "Page 3, line 24". Here, the "weather stations" imply the real weather stations (i.e., ASOS + AWS + AAOS; see Fig. R1). As mentioned in (1), we used the weather station data to classify the precipitaion types. The base data for classification is shown in Table R1 below (not included in the revised manuscript). The 1 km composite data are produced using the radar, station and satellite data, through the method described in Hwang et al. (2015) — see also the step-by-step description below. We have rewritten the first and second paragraphs in Sec. 2, with new statements in bold, and reorganized it as:

**We used the precipitation data from weather stations, shown in Fig. 1, to categorize the precipitation systems.** We classify four different precipitation types statistically based on two criteria: the portion of weather stations with precipitation (C1), and

the station average precipitation rate (C2). Based on these criteria, we define four different precipitation types, as shown in Table 1: 1) Low Precipitation at a Few Points (LPFP) for C1 $< 20$ % and C2 $< 3$ mm h$^{-1}$; 2) Low Precipitation at Many Points (LPMP) for C1 $\geq 20$ % and C2 $< 3$ mm h$^{-1}$; 3) High Precipitation at a Few Points (HPFP) for C1 $< 20$ % and C2 $\geq 3$ mm h$^{-1}$; and 4) High Precipitation at Many Points (HPMP) for C1 $\geq 20$ % and C2 $\geq 3$ mm h$^{-1}$. We practically exclude the LPFP type in our analyses, i.e., the case with C1 $< 20$ % and C2 $< 3$ mm h$^{-1}$, because it may be less effective.

The Korea Meteorological Administration (KMA) has produced a composite precipitation data over Korea **using the data from radars, weather stations and satellites, through the following steps as described in Hwang et al. (2015): 1) remove non-precipitation echoes from the radar data using the satellite cloud type data; 2) calculate the difference between the station precipitation and the radar estimated precipitation; 3) perform the objective analysis on the precipitation difference field and on the station precipitation data; 4) correct the bias using the objectively-analyzed difference field; and 5) combine the corrected radar-estimated precipitation data and the objectively-analyzed station precipitation data to produce the composite precipitation data (in mm h$^{-1}$). In order to analyze the precipitation systems with high resolution and evenly distributed data, we used this composite precipitation data.** This data covers 1153 km $\times$ 1441 km over the Korean Peninsula, with a grid size of 1 km and a time resolution of 1 h. Geostatistical analyses are conducted using this composite precipitation data sets from April to October in a period of 2013–2015 to investigate the spatial and temporal characteristics of summer rainfall.

(3) *Provide the formulation of the weighting function $w_{ij}(d)$ in Eq. (7) and explain why the specific form is selected to represent the spatial variations.*

$\Longrightarrow$ We used an inverse distance weighting (IDW) function, i.e., $w_{ij}(d) = 1/d_{ij}$ where $d_{ij}$ is the distance between grids $i$ and $j$. In fact, this is the same as the one used in calculating Moran's $I$ (see Eqs. (4) and (5)). The IDW is a widely-used one among the spatial weighting functions. We have modified the sentence that defines $w_{ij}$ (page 5, line 20, below Eq. (4), in the original manuscript) as (the modified parts in bold):

"$\cdots$ Here, $w_{ij}$ is the spatial weight of the link between $i$ and $j$, which is defined by the inverse distance weight**, i.e.,** $w_{ij} = 1/d_{ij}$ **with** $d_{ij}$ **representing the distance between grids** $i$ **and** $j$. $\cdots$"

We have also modified the expression right below Eq. (7) as (the modified parts in bold):

"where $d$ is the distance between the target feature and the neighboring feature**, and** $w_{ij}$ **is the same spatial weight (i.e., the inverse distance weight) used for calculating Moran's** $I$ **as in Eqs. (4) and (5).** $\cdots$"

(4) *Page 7, line 20: This result is trivial consequence of classifying the precipitation system in terms of a number of data points; if rainfall occurs only over a small number of points, its spatial scale is limited by design. Their selection of 3 mm/h as the threshold value between heavy and light precipitation may cause the lack of relationship between precipitation intensity and spatial scales. Nam et al. may be a good reference for this. In fact, this result can depend on the selection of the threshold value. The authors need to explain the choice of 3 mm/h as the threshold.*

$\Longrightarrow$ The threshold values (i.e., 20% and 3 mm/h) were taken based on a preliminary statistical analysis of precipitation events, as shown in Table R1. Since

we are dealing with precipitation occurrences for a given time period and/or space interval, our data mostly follow the Poisson distribution. In classifying the precipitation types, we used two criteria — the portion of weather stations with precipitation and the station average precipitation rate — and determined the threshold values when the cumulative percentage of each criterion reaches 80% (see the red lines in Table R1). In terms of the portion of weather stations with precipitation, the cumulative percentage reaches 77.1% with the portion of 10–20% and 85.0% with the portion of 20–30%; thus selecting 20% as the threshold value. In terms of the station average precipitation rate, the cumulative percentage becomes 80.0% with 2.0–2.9 mm/h and 93.3% with 3.0–4.9 mm/h; thus choosing 3 mm/h as the threshold value. As we have selected 3 mm/hr as the threshold value based on this statistical analysis, this selection may not cause a lack of relationship between precipitation intensity and spatial scales. Actually Table R1 shows that heavy precipitation systems have high locality, which is consistent with the findings of Nam et al. (2014). Especially precipitation with the highest intensity ($\geq 10$ mm/hr) mostly occurs in a small area with the number of stations less than 10% of total weather stations. We have added the following statement at the early part of Sec. 2 in the revised manuscript to describe the background of selecting the threshold values. We have also addressed the referee's point about possible dependency of results on the threshold value.

In order to determine the threshold values for classifying the precipitation types, we have conducted a preliminary statistical analysis on precipitation events in the period of 2011–2015 (not shown). As the precipitation events occur in a given time period and/or space interval, our precipitation data are assumed to follow the Poisson distribution, which represents a probability situation of a large number of observation with a small probability of occurrence.

Many studies developed the Poisson distribution models to estimate rainfall and cluster the rainfall systems (e.g., Rodriguez-Iturbe et al., 1987; Lee et al., 2014; Barton et al., 2016; Ritschel et al., 2017). We have chosen the threshold values when the cumulative percentage of precipitation events for each criterion (i.e., C1 and C2) reached approximately 80%; thus obtaining the threshold values of 20% for C1 and 3 mm h$^{-1}$ for C2, respectively. Our preliminary statistical analysis showed that, in general, most precipitation events occur over small areas and precipitation events with high intensity rarely occur over large areas. The locality of precipitation appeared higher as the precipitation intensity were higher, in accordance with Nam et al. (2014). In particular, precipitation systems with the highest intensity ($\geq 10$ mm/hr) were mostly confined to a small area with the number of stations less than 10% of total weather stations. This implies that the locality feature of precipitation systems may depend on the threshold value in precipitation intensity.

(5) *Page 7, line 22 – Page 8, line 15: The authors need to clearly state how the analyses in this block are related to the "propagation of precipitation systems". Analyses in this block are directly related to spatial structures (e.g., shape and orientation); please explain how can these features be related to "propagation".*

$\implies$ We appreciate the referee pointing this out. We noticed that it may cause confusion because the meaning of "propagate" includes "to travel through space" (Merriam-Webster), which entails the concept of time dimension. We actually used the word "propagation" because all the weather systems evolve in time (i.e., develop, mature and decay) and move in space during their life cycles. Furthermore, as we used the 3-year data, the spatial auto-correlations here are considered to include the temporal features implicitly. The fact that precipitation systems have high correlation along a specific direction implies that those systems in that direction, even in far distances, have similar/common structures or are originated from the same weather system. For the meteorological systems with strong directionality, we can mention a squall line or a frontal system in which several thunderstorms banded together, and a multicell cluster that includes a series of individual storm at a different stage of life cycle with the same movement direction. For the multicell cluster, new cells form along the upwind edge of the cluster, and decaying cells are found along the downwind side with mature cells located at the center; thus it includes evolutions in both space and time. We addressed this point in the revised manuscript; nevertheless, we decided to modify the expression "the spatial **propagation** of precipitation systems" to "the spatial **structures** of precipitation systems" (page 7, line 25), to avoid any confusion.

(6) *The spatial shape differences between the three rainfall types (in the same block as above): the asymmetry indicated in the spatial correlation (Fig. 4) does not correspond well to that depicted in the radar echo (Fig. 5). The directional difference in the e-folding scale for the all three systems are about 25% (5km/20km for HPFP; 10km/40km for HPMP and LPMP) of the mean scale (i.e., aspect ratios of ∼1.3) while the radar echoes suggest larger aspect ratios for HPFP and HPMP (∼1). This is not consistent with their interpretation of rainfall system in Page 8, line 10: how often a squall line is of an aspect ratio of 2?*

⟹ We are not sure if we have fully understood the referee's point/question here. In our understanding, the directional difference of 25% seems to indicate the ratio of the difference between the largest and the smallest values of the directional $e$-folding distances (say, $\Delta d$) and the largest directional $e$-folding distance (say, $d_{max}$), i.e., $\Delta d/d_{max}$. For example, in Fig. 4b, LPMP shows $\Delta d \simeq 10$ km and $d_{max} \simeq 40$ km; thus making the ratio be $\sim 0.25 = 25\%$. It seems that the referee defined the aspect ratio as the largest directional

$e$-folding distance divided by the smallest directional $e$-folding distance (say, $d_{min}$), i.e., $d_{max}/d_{min}$. For example, in Fig. 4b, LPMP shows $d_{max} \simeq 40$ km and $d_{min} \simeq 30$ km; thus making the aspect ratio be $\sim 1.3$. Probably the referee wanted to mean the aspect ratio of HPFP and HPMP in the radar diagram (Fig. 5) to be $\sim 2$, not $\sim 1$.

$\Longrightarrow$ If our understanding above is correct, we make the following reply to the referee's comment. Figure 4 shows the mean directional autocorrelation for different precipitation types, whereas Fig. 5 shows the directional $e$-folding distance regarding to all cases in each precipitation type by finding the mode in the histogram. Note that the mean $e$-folding distance (Fig. 4) and the $e$-folding distance of the mode (Fig. 5) do not necessarily be the same or similar. Although Figs. 4 and 5 are based on the same composite precipitation data, they may be different especially when the deviation of $e$-folding distance from each case is large. Therefore, they may not match each other for the $e-$folding distances and hence the scales of spatial correlation.

$\Longrightarrow$ Furthermore, please note that the radar diagram (Fig. 5) does not mean the diagram representing radar echoes as the referee mentioned. In other words, Fig. 5 is not related to Fig. 4 in any aspect. To avoid any confusion, we have modified the "radar diagram" to the "radar chart" in the revised manuscript. A *radar chart*, also known as web chart or spider chart, is a graphical method of displaying multivariate data in the form of a two-dimensional chart of three or more quantitative variables represented on axes starting from the same point (source: en.wikipedia.org).

$\Longrightarrow$ We have mentioned that our HPMP case "may" correspond to squall lines as well as convection bands, cloud clusters or the warm-type heavy rainfall, based on the previous studies. However, we have not conducted the analysis for each precipitation type (i.e., squall lines, convection bands, etc.), and hence we do not have any information on the aspect ratio of squall lines in Korea. It is essential to conduct the case studies for each precipitation type

in the future.

(7) *Page 8: If the satellite data cannot clearly distinguish the areas of water vapor from those without, how much can we trust the analysis based on the data? Can they provide data quality control of the satellite data?*

⟹ Satellite images from the water vapor (WV) channels represent several important dynamical features in the upper- and mid-level atmosphere: dry air or clear sky is represented by dark area, moist air or cloudy sky by light area, intrusion of dry air in mature cyclones by dry slot (i.e., dark area), jet stream location by high contrast between dark and light areas (i.e., boundary), upper-level vorticity by rotational patterns of light and dark areas, and so on. In particular, the two WV channels 6.2 and 7.3 $\mu$m are sensitive to catch moisture boundaries at the zone between the warm/moist and cold/dry side of the jet/wind maximums at two different levels in the troposphere (Georgiev et al., 2016). Therefore, the satellite data from the WV channels clearly distinguish the dry vs. humid air, and detect the moisture boundary effectively. In the text, we mentioned that it was hard to distinguish between water vapor and clouds, thus used the mixed images. Given that both WV and clouds are the sources of precipitation, analysis of the mixed variables from the satellite data may not make a serious problem in understanding and relating to the precipitation systems. We have rewritten this part in the revised manuscript to deliver our intent more clearly (see the second paragraph of Sec. 4.1.2) as (new sentences in bold):

> In this study, we analyze the brightness temperatures from the Himawari-8 water vapor bands to characterize the lower to upper atmosphere related to the precipitation systems. **A humid atmosphere absorbs more longwave radiation from the Earth, resulting in a lower brightness temperature. On the other hand, a dry atmosphere absorbs less longwave radiation, bringing**

**about a higher brightness temperature. Although we cannot directly quantify the amount of water vapor through the water vapor imager, we can sufficiently recognize the spatial distribution of water vapor. Moreover, using two water vapor channels (i.e., 6.2 and 7.3 $\mu$m), we can clearly identify the moisture boundaries at the zone between the warm/moist and cold/dry side of the jet/wind maximums at two different levels in the troposphere (Georgiev et al., 2016).** The spatial analyses were performed with the mixed images of clouds and water vapors because it was hard to distinguish between clouds and water vapor without a cloud detection algorithm. **Since both water vapor and clouds are strongly linked to precipitation as its sources, analysis of the mixed variables from the satellite data would not make a serious problem in understanding and relating to the precipitation systems.** As we focus on the spatial distribution of water vapor when precipitation occurs, we analyze water vapor for each precipitation type.

$\Longrightarrow$ Furthermore, we have added the description of the satellite data quality with new references (i.e., Okuyama et al., 2015) to Sec. 2 as:

> The calibration of the Himawari-8 water vapor bands is accurate to within 0.2 K by validating an approach developed under the Global Space-based Inter-calibration System (GSICS) project with hyperspectral infrared sounders (e.g., Okuyama et al., 2015; Bessho et al., 2016).

(8) *Page 8, line 33: The only similarity between Fig. 2 and Fig. 6 is that the autocorrelation for HPFP decreases more rapidly than those for HPMP and LPMP. The separation between HPFP and HPMP/LPMP in Fig. 6 is much smaller than in Fig. 2 as well. Overall, it's difficult to establish similarity between the spatial scales of*

*water vapor and rainfall. The authors need to provide clear explanations on how to related the structures based on water vapor scales (Fig. 6) to that based on rainfall (Fig. 2). Overall, it is difficult to much merits of the satellite vapor analysis towards the rainfall structure over Korea.*

$\Longrightarrow$ We agree with the referee that it is difficult to establish similarity between the spatial scales of water vapor and rainfall. However, our intent on analyzing the satellite WV data was not only to establish similarity but also to find any possible dissimilarity between WV and precipitation. Since the WV makes phase changes and the conversion from WV to precipitation includes a bunch of nonlinear processes, we did not expect high similarity in the spatial scales and structure between WV and precipitation. Rather, we had scientific curiosity on the degree of similarity vs. dissimilarity, and on what aspects of dissimilarity would be found. As we have focused on the comparison of spatial correlation among precipitation types, we found the similarity between precipitation (Fig. 2) and water vapor (Fig. 6), as mentioned by the referee: the autocorrelation of WV for HPFP decreases more rapidly than those for HPMP and LPMP as in precipitation. We have also found and discussed the dissimilarity, especially on the separation distance, i.e., the spatial scales. We additionally discovered the similarity between WV and precipitation in terms of characteristic directionality in spatial autocorrelations. Through the satellite WV analyses, we also aimed at examining the degree and direction of spatial correlation of WV into the Korean Peninsula in association with the precipitation types. In addition, many studies have been done in the past about the relationship between satellite water vapor and extratropical/tropical cyclones and storms (e.g., Velden, 1987; Milford and Dugdale, 1990; Krennert and Zwatz-Meise, 2003; De Haan et al., 2004; Rabin et al., 2004; Mukhopadhyay et al., 2005; Chosh et al., 2008). Further studies on the relationship between the precipitation systems and satellite

water vapor in Korea are essentially required.

$\implies$ We have addressed these points in the revised manuscript. We first modified the first paragraph of Sec. 4.1.2, with new sentences in bold, as:

> Water vapor is the core element and driver of the precipitation development through dynamical processes (e.g., advection and convection) and physical processes (e.g., evaporation and condensation). For example, the East Asian monsoon starts when a huge amount of water vapor from the adjacent ocean is transported to the monsoon region by the large scale atmospheric circulation. Thus, the spatial analysis of water vapor will contribute to further understanding of the spatial patterns of precipitation. **Many studies have been done in the past about the relationship between satellite water vapor and extratropical/tropical cyclones and storms (e.g., Velden, 1987; Milford and Dugdale, 1990; Krennert and Zwatz-Meise, 2003; De Haan et al., 2004; Rabin et al., 2004; Mukhopadhyay et al., 2005; Chosh et al., 2008).**
>
> By analyzing the water vapor imagery, we can detect not only the horizontal distribution of tropospheric water vapor but also the dynamical behavior of atmospheric flow such as the middle and upper troughs, vortexes, and jet streams, even in the absence of clouds. Therefore, analyzing the spatial characteristics of tropospheric water vapor is essential to improve the understanding of precipitation systems. **In particular, we are focusing on the degree and direction of spatial correlation of water vapor into the Korean Peninsula in association with the precipitation types.**

We have also added the following paragraph to the end of Sec. 4.1.2:

> Through the satellite water vapor analyses, we found both similarity and dissimilarity in spatial correlations between water vapor

and precipitation. Similar to the results of precipitation analyses, the spatial autocorrelations of water vapor for HPFP decreased more rapidly than those for HPMP and LPMP (cf. Figs. 2 and 6). Water vapor and precipitation showed additional similarity in terms of characteristic directionality in spatial autocorrelations (cf. Figs. 4 and 7, and Figs. 5 and 8). However, both fields showed significant dissimilarity in the separation distances, i.e., the spatial scales (cf. Figs. 2 and 6). Note that water vapor makes phase changes and the conversion from water vapor to precipitation includes a bunch of nonlinear processes; thus it is not surprising to see such dissimilarity. Further studies on the relationship between the precipitation systems and satellite water vapor in Korea are essentially required.

(9) *Temporal correlation analysis: It's not clear what we can learn about the rainfall systems from the temporal correlation characteristics. The $e$-folding scale differs by only 30 mins among the three types. The time scale of 1–1.5 hours seem to indicate that the all three rainfall types are related to convective systems, either isolated or clustered. Does this provide any insights to separate the characteristics of the three rainfall types? Also, it's not clear how the water vapor analysis can be related to the rainfall characteristics.*

⟹ We appreciate the referee pointing this issue out. We agree with the referee that the $e$-folding time is in the range of 1–2 hours and there is no significant difference by precipitation types. Nevertheless, we believe that this finding has some meaningful implications. First, as the referee has pointed out, the relatively short $e$-folding time scale for all precipitation types implies that the precipitation systems affecting Korea in summer are mostly characterized by convective-type precipitation, from either isolated storm cells or clustered bands, at least for the analysis period. Second, the $e$-folding time scale suggests a proper time interval for data collection and analysis for capturing the

detailed structure of and better forecasting of precipitation systems. Lastly, it has another important implication on data assimilation, especially on the proper time interval of incorporating observations in the operational data assimilation system, for more accurate numerical forecasting of the precipitation systems. We also note that a similar result was reported by Ha et al. (2007), showing the $e$-folding time of precipitation in Korea is 1–2 hours regardless of months (i.e., May to September). Therefore, we can conclude that the typical $e$-folding time of the precipitation systems in Korea is 1–2 hours, regardless of the precipitation types. This conclusion is based on the analyses of the hourly precipitation data: we may find different temporal characteristics for different precipitation types using a data set of shorter interval (e.g., 10 min). A further work is also necessary to investigate the relationship between the temporal scale as well as spatial scale of water vapor transport and precipitation with more detailed analysis. We have addressed these points in the revised manuscript by adding the following two paragraphs to the end of Sec. 4.3:

Through this temporal correlation analyses, we noticed that the $e$-folding time is in a short range (1–2 hours), and its difference among different precipitation types is only about 30 min. This short $e$-folding time scale for all precipitation types implies that the precipitation systems affecting Korea in summer are mostly characterized by convective-type precipitation, from either isolated storm cells or clustered bands, at least for the given analysis period. The $e$-folding time scale suggests an adequate time interval for data collection and analysis for capturing the detailed structure of and better forecasting of precipitation systems. It also implies a proper time interval of incorporating observations in the operational data assimilation system, for more accurate numerical forecasting of the precipitation systems.

Moreover, Ha et al. (2007) reported that the $e$-folding time of precipitation in Korea is 1–2 h regardless of months (from May to September), and that the monthly difference of the $e$-folding time is approximately 30–40 min. Therefore, we can conclude that the typical $e$-folding time of the summer precipitation systems in Korea is 1–2 h, regardless of the precipitation types and months. This conclusion is based on the analyses of the hourly precipitation data: we may find different temporal characteristics for different precipitation types using a data set of shorter interval (e.g., 10 min). In terms of the satellite water vapor data, a further work is also necessary to investigate the relationship between the temporal scale as well as spatial scale of water vapor transport and precipitation with more detailed analysis.

(10) *Considering the aspect ratio and spatial scales, the examples in Fig. 11 seem more relevant for convective clusters (may be imbedded within a frontal structure) than a frontal system.*

$\implies$ As Fig. 11 depicts an HPFP case in October, we agree with the referee's opinion that the case seems to be related to convective clusters. In Section 4.1.1, we also mentioned that HPFP may correspond to the isolated thunderstorms or convection bands as well as frontal rainfalls. To avoid confusion, we have rewritten the statement in page 10, line 25 as:

"For example, precipitation in the HPFP case**, relevant to convective clusters,** appeared over a small local area (Fig. 11a) but the value of global Moran's $I$ was much higher (0.2357) than the others (0.1207 and 0.1711), implying a stronger cluster pattern; however, considering that the global Moran's $I$ is a domain-averaged value, this high value may be due to less dispersion (negative correlation) areas."

**Table R1.** Preliminary statistical analysis of precipitation events during 2011–2015 by two criteria —- the portion of weather stations with precipitation and the station average precipitation rate. The red lines indicate the boundaries when the cumulative percentage of precipitation events is approximately 80 %.

**Figure R1.** The weather station locations (blue dots) and radar locations (red plus symbols) and coverages (white area) in Korea.

**References**

Barton, Y., Giannakaki, P., Von Waldow, H., Chevalier, C., Pfahl, S., and Martius, O.: Clustering of regional-scale extreme precipitation events in Southern Switzerland, Mon. Wea. Rev., 144, 347–369, 2016.

Bessho, K., Date, K., Hayashi, M., Ikeda, A., Imai, T., Inoue, H., · · ·, and Yoshida, R.: An introduction to Himawari-8/9 — Japan's new-generation geostationary meteorological satellites, J. Meteorol. Soc. Japan Ser. II, 94, 151–183, 2016.

Choi, S.-W., Lee, S.-J., Kim, J. Lee, B.-L., Kim, K.-R., and Choi, B.-C.: Agrometeorological observation environment and periodic report of Korea Meteorological Administration: Current status and suggestions, Korean J. Agric. For. Meteorol., 17, 144–155, doi:10.5532/KJAFM.2015.17.2.144, 2015 (in Korean with English abstract).

Chosh, A., Lohar, D., and Das, J.: Initiation of Nor'wester in relation to mid-upper and low-level water vapor patterns on METEOSAT-5 images, Atmos. Res., 87, 116–135, 2008.

De Haan, S., Barlag, S., Baltink, H. K., Debie, F., and Van der Marel, H.: Synegetic use of GPS water vapor and Meteosat images for synoptic weather forecasting, J. Appl. Meteor., 43, 514–518, 2004.

Georgiev, C. G., Santurette, P., and Maynard K.: Weather Analysis and Forecasting: Applying Satellite Water Vapor Imagery and Potential Vorticity Analysis, 2nd Ed., Academic Press, Cambridge, USA, 2016.

Krennert, T. and Zwatz-Meise, V.: Initiation of convective cells in relation to water vapour boundaries in satellite images, Atmos. Res., 67–68, 353–366, 2003.

Lee, J., Yoon, J., and Jun, H. D.: Evaluation for the correction of radar rainfall due to the spatial distribution of raingauge network, J. Korea Soc. Hazzard Mitig., 14, 337–345, http://dx.doi.org/10.9798/KOSHAM.2014.14.6.337, 2014.

Milford, J.R. and Dugdale, G.: Estimation of rainfall using geostationary satellite data, in: Applications of Remote Sensing in Agriculture, Steven, M. D. and Clark, J. A., Eds., Butterworth, 97–110, 1990.

Mukhopadhyay, P., Singh, H. A. K., and Singh, S. S.: Two severe Nor'westers in April 2003 over Kolkata using Doppler Radar observations and satellite imagery, Weather, 60, 343–353, 2005.

Nam, J.-E., Lee, Y. H., Ha, J.-C., and Cho, C.-H.: A study on the e-folding distance of summer precipitation using precipitation reanalysis data, in: Proceedings of the Autumn Meeting of Korean Meteorological Society, 2014, Korean Meteorol. Soc., Jeju, Korea, 13–15 October 2014, 657–658, 2014.

Okuyama, A., Andou, A., Date, K., Hosaka, K., Mori, N., Murata, H., Tabata, T., Takahashi, M., Yoshino, R. and Bessho, K.: Preliminary validation of Himawari-8/AHI navigation and calibration. Proc. SPIE 9607, Earth Observing Systems XX, 96072E, doi:10.1117/12.2188978, 2015.

Rabin, R. M., Corfidi, S. F., Brunner, J. C., and Hane, C. E.: Detecting winds aloft from water vapor satellite imagery in the vicinity of storms, Weather, 59, 251–257, doi:10.1256/wea.182.03, 2004.

Ritschel, C., Ulbrich, U., Névir, P., and Rust, H. W.: Precipitation extremes on multiple timescales — Bartlett-Lewis rectangular pulse model and intensity-duration-frequency curves, Hydrol. Earth Syst. Sci., 21, 6501–6517, https://doi.org/10.5194/hess-21-6501-2017, 2017.

[Figure]

Rodriguez-Iturbe, I., Cox, D. R., and Isham, V.: Some models for rainfall based on stochastic point processes, Proc. R. Soc. London, Ser. A, 410, 269–288, 1987.

Velden, C. S.: Satellite observations of Hurricane Elena (1985) using the VAS 6.7 $\mu$m "water-vapor" channel, Bull. Amer. Meteor. Soc., 68, 210–215, 1987.

[Figure]

| The station average precipitation rate (mm/h) | The portion of weather stations with precipitation (%) | | | | | | | | | | Sum | Percentage (%) | Cumulative percentage (%) |
|---|---|---|---|---|---|---|---|---|---|---|---|---|---|
| | 0-10 | 10-20 | 20-30 | 30-40 | 40-50 | 50-60 | 60-70 | 70-80 | 80-90 | 90- | | | |
| 0.1-0.9 | 5241 | 399 | 156 | 60 | 29 | 5 | 2 | 1 | 0 | 0 | 5893 | 40.0 | 40.0 |
| 1.0-1.9 | 2238 | 665 | 383 | 235 | 148 | 99 | 57 | 33 | 18 | 6 | 3882 | 26.3 | 66.3 |
| 2.0-2.9 | 891 | 352 | 262 | 196 | 109 | 89 | 48 | 35 | 29 | 12 | 2023 | 13.7 | 80.0 |
| 3.0-4.9 | 765 | 322 | 224 | 212 | 159 | 85 | 77 | 43 | 49 | 35 | 1971 | 13.4 | 93.3 |
| 5.0-9.9 | 317 | 118 | 116 | 115 | 76 | 47 | 35 | 19 | 18 | 8 | 869 | 5.9 | 99.2 |
| 10.0- | 61 | 6 | 19 | 11 | 14 | 1 | 0 | 0 | 0 | 0 | 112 | 0.8 | 100.0 |
| Sum | 9513 | 1862 | 1160 | 829 | 535 | 326 | 219 | 131 | 114 | 61 | 14750 | | |
| Percentage (%) | 64.5 | 12.6 | 7.9 | 5.6 | 3.6 | 2.2 | 1.5 | 0.9 | 0.8 | 0.4 | | | |
| Cumulative percentage (%) | 64.5 | 77.1 | 85.0 | 90.6 | 94.2 | 96.4 | 97.9 | 98.8 | 99.6 | 100.0 | | | |

**Fig. 1.** Table R1. See the caption for Table R1 in C17.

**Fig. 2.** Figure R1. See the caption for Figure R1 in C17.